# Prophage regulation of *Shewanella fidelis* 3313 motility and biofilm formation with implications for gut colonization dynamics in *Ciona robusta*

Ojas Natarajan[1], Susanne L Gibboney[1], Morgan N Young[1], Shen Jean Lim[2], Felicia Nguyen[3], Natalia Pluta[3†], Celine GF Atkinson[4‡], Assunta Liberti[1,5], Eric D Kees[6], Brittany A Leigh[1,2§, #], Mya Breitbart[2], Jeffrey A Gralnick[6,7], Larry J Dishaw[1,3,4*]

[1]Department of Pediatrics, Morsani College of Medicine, University of South Florida, Tampa, United States; [2]College of Marine Science, University of South Florida, Tampa, United States; [3]Biomedical Sciences Program, University of South Florida, Tampa, United States; [4]Department of Cell Biology, Microbiology, and Molecular Biology, University of South Florida, Tampa, United States; [5]Stazione Zoologica Anton Dohrn, Naples, Italy; [6]BioTechnology Institute, University of Minnesota, St. Paul, United States; [7]Plant and Microbial Biology, University of Minnesota, St. Paul, United States

**\*For correspondence:**
ldishaw@usf.edu

**Present address:** [†]Uniformed Services University of Health Sciences, Bethesda, United States; [‡]School of Dentistry, University of Washington, Seattle, United States; [§]Cargill Corporation, Wayzata, United States; [#]LifeSci Communications, New York and Boston, United States

**Competing interest:** The authors declare that no competing interests exist.

## eLife Assessment

This **valuable** study presents findings linking prophage carriage to lifestyle regulation in the marine bacterium *Shewanella fidelis*, with potential implications for niche occupation within a host (*Ciona robusta*) and mediation of host immune responses. The study leverages a unique animal model system that offers distinct advantages in identifying select phenotypes to present overall **solid** evidence that supports findings relating to the impact of a prophage on host-microbe interaction. Understanding the role of integrated lysogenic phages in bacterial fitness, both within a host and in the environment, is a significant concept in bacterial eco-physiology, potentially contributing to the success of certain strains.

**Abstract** Lysogens, bacteria with one or more viruses (prophages) integrated into their genomes, are abundant in the gut of animals. Prophages often influence bacterial traits; however, the influence of prophages on the gut microbiota–host immune axis in animals remains poorly understood. Here, we investigate the influence of the prophage SfPat on *Shewanella fidelis* 3313, a persistent member of the gut microbiome of the model marine tunicate, *Ciona robusta*. Establishment of a SfPat deletion mutant (ΔSfPat) reveals the influence of this prophage on bacterial physiology in vitro and during colonization of the *Ciona* gut. In vitro, deletion of SfPat reduces *S. fidelis* 3313 motility and swimming while increasing biofilm formation. To understand the in vivo impact of these prophage-induced changes in bacterial traits, we exposed metamorphic stage 4 *Ciona* juveniles to wildtype (WT) and ΔSfPat strains. During colonization, ΔSfPat localizes to overlapping and distinct areas of the gut compared to the WT strain. We examined the differential expression of various regulators of cyclic-di-GMP, a secondary signaling molecule that mediates biofilm formation and motility. The *pdeB* gene, which encodes a bacterial phosphodiesterase known to influence biofilm formation and motility by degrading cyclic-di-GMP, is upregulated in the WT strain but not in ΔSfPat when examined in vivo. Expression of the *Ciona* gut immune effector, VCBP-C, is enhanced during colonization

by ΔSfPat compared to the WT strain; however, VCBP-C binding to the WT strain does not promote the excision of SfPat in an SOS-dependent pathway. Instead, VCBP-C binding significantly reduces the expression of a phage major capsid protein. Our findings suggest that SfPat influences host perception of this important colonizing commensal and highlights the significance of investigating tripartite dynamics between prophages, bacteria, and their animal hosts to better understand the gut microbiota-host immune axis.

## Introduction

An epithelial layer with a mucus-rich surface lines the gastrointestinal tract (or gut) of animals. This dynamic environment is a primary interface between host immunity, symbiotic microbes, and dietary antigens (*Li et al., 2015*; *Johansson et al., 2011*). During colonization of the gut, bacteria encounter physical, chemical, and biological forces. They must compete for nutrients and niche space while managing the stress of digestive enzymes and host immune factors (*Duncan et al., 2021*; *Harrington et al., 2021*; *Hornung et al., 2018*; *Moran et al., 2019*). Gut-colonizing bacteria also encounter bacteriophages (phages), or viruses that infect bacteria (*Mirzaei and Maurice, 2017*); for example, over $10^{12}$ viruses have been estimated in the human gut (*Shkoporov and Hill, 2019*).

Phages display both lytic and temperate lifestyles. While the former impacts bacterial community dynamics through lysis, the effect of temperate phages on bacterial communities, especially those associated with animals, remains poorly understood. Conventionally, temperate phages integrate into bacterial genomes as prophages and remain 'dormant' until an external trigger activates them to enter the lytic cycle (*Lwoff, 1953*; *Boling et al., 2020*; *Howard-Varona et al., 2017*); these bacteria are considered 'lysogenized' and referred to as lysogens (*Lwoff, 1953*; *Howard-Varona et al., 2017*). Prophages often encode accessory genes that can influence bacterial traits and behaviors (*Mills et al., 2013*). These genes can encode virulence factors, antibiotic resistance determinants, and those that provide superinfection exclusion, thereby protecting their bacterial hosts from infections by related phages (*Bondy-Denomy and Davidson, 2014*). Based on the site of integration, prophages can also impact the expression of bacterial genes (*Aziz et al., 2005*). Bacterial lysogens exist in every environment (*Jiang and Paul, 1998*; *Silveira et al., 2021*; *Leigh et al., 2018*) and are particularly prevalent in the microbiomes of diverse animals (*Kim and Bae, 2018*; *Shkoporov and Hill, 2019*).

Prophage induction results in bacterial lysis and can be mediated by various stressors, including antibiotics and inflammatory processes (*Allen et al., 2011*; *Banks et al., 2003*; *Diard et al., 2017*; *Fang et al., 2017*; *Garcia-Russell et al., 2009*; *Maiques et al., 2006*; *Nanda et al., 2015*; *Wang et al., 2010*; *Zhang et al., 2000*). Because prophages can influence the phenotypes of their bacterial hosts, such as biofilm formation (*Nanda et al., 2015*), understanding the impact of lysogens in animal microbiomes is becoming a research priority (*Fortier and Sekulovic, 2013*; *Hu et al., 2021*; *Lin et al., 1999*). In the gut, biofilms that associate with host mucus may benefit the host by enhancing epithelial barriers against pathogenic invasion (*Swidsinski et al., 2007*). Integration of prophages into bacterial genomes may impart functional changes that could be important in surface colonization. For example, in *Escherichia coli* K-12, integrating the Rac prophage into a tRNA thioltransferase region disrupts biofilm functions (*Liu et al., 2015*); deleting this prophage decreases resistance to antibiotics, acids, and oxidative stress. Some of these traits may be affected by prophage-specific genes (*Wang et al., 2010*). Prophages have also been shown to influence biofilm life cycles in *Pseudomonas aeruginosa* (*Rice et al., 2009*).

Lysogenized *Shewanella* species colonize the gut of *Ciona robusta*, an ascidian collected in Southern California waters (*Leigh et al., 2017*) and referred to here as *Ciona*. Tunicates like *Ciona* are a subphylum (Tunicata) of chordates that are well-established invertebrate model systems for studies of animal development (*Chiba et al., 2004*; *Davidson, 2007*; *Liu et al., 2006*) and are now increasingly leveraged for gut immune and microbiome studies (*Liberti et al., 2021*). Armed with only innate immunity, *Ciona* maintains stable gut bacterial and viral communities (*Dishaw et al., 2014b*; *Leigh et al., 2018*) despite continuously filtering microbe-rich seawater. Previous efforts to define gut immunity in *Ciona* revealed the presence of a secreted immune effector, the variable immunoglobulin (V-Ig) domain-containing chitin-binding protein, or VCBP, that likely plays important roles in shaping the ecology of the gut microbiome by binding bacteria and fungi (as well as chitin-rich mucus) on opposing functional domains (*Dishaw et al., 2011*; *Liberti et al., 2019*). VCBP-C is one of

the best-studied VCBPs expressed in the stomach and intestines of *Ciona*, shown to bind bacteria in vitro and in the gut lumen (*Dishaw et al., 2016*; *Dishaw et al., 2011*). Based on various in vitro and in vivo observations, it was proposed previously that VCBPs likely modulate bacterial settlement and/ or biofilm formation (*Dishaw et al., 2016*; *Liberti et al., 2019*; *Liberti et al., 2021*). The potential influence of soluble immune effectors on host–bacterial–viral interactions is particularly interesting. However, the possibility that prophages may influence interactions between bacteria and secreted immune effectors like VCBPs remains to be explored.

Shewanella fidelis strain 3313 was isolated previously from the gut of *Ciona* and found to possess two inducible prophages, SfPat and SfMu *Leigh et al., 2017*. Furthermore, in vitro experiments demonstrated enhanced biofilm formation in *S. fidelis* 3313 in the presence of extracellular DNA (eDNA) that may originate from lytic phage activity (*Leigh et al., 2017*). Other *Shewanella* species have previously demonstrated a link between phage-mediated lysis and biofilm formation (*Gödeke et al., 2011*). For example, in *S. oneidensis* strain MR-1, Mu and Lambda prophages enhance biofilm formation via eDNA released during prophage-induced lysis, with genomic DNA likely serving as a scaffold for biofilms (*Gödeke et al., 2011*). Similarly, the P2 prophage of *S. putrefaciens* strain W3-18-1 influences biofilm formation via spontaneous induction at low frequencies, resulting in cell lysis and contributing eDNA that can mediate biofilm formation (*Liu, 2019*).

Here, we set out to isolate and characterize the influence of the prophage, SfPat, on its host, *S. fidelis* 3313. Since the last description (*Leigh et al., 2017*), the genome of *S. fidelis* 3313 was improved by combining long-read and short-read sequencing, which resulted in improved resolution of the genomic landscape within and around the prophage. A homologous recombination-based deletion strategy was designed to generate a deletion mutant (i.e., knockout) of SfPat. We report that deletion of SfPat results in reduced bacterial motility and increased biofilm formation in vitro. These changes in bacterial traits and behaviors are associated with the expression of genes regulating important signaling molecules and a corresponding impact on host immune gene expression during gut colonization in *Ciona* juveniles. Gut colonization experiments in laboratory-reared *Ciona* juveniles comparing wild-type (WT) and SfPat prophage knockout (ΔSfPat) mutant strains demonstrate that SfPat influences gut colonization outcomes, for example, niche preference and retention. These effects are influenced by host gene expression. The results reported herein reflect complex tripartite interactions among prophages, bacterial hosts, and animal immune systems.

## Results

### Sequence verification of prophage deletion mutant strains

Colony PCR and single primer extension sequencing were both used to validate the prophage deletion, using primers EDK81/82 for SfPat (*Figure 1a*). All recovered amplicon sizes were consistent with the predictions for SfPat deletion (*Figure 1b*). The SfPat deletion (ΔSfPat) strain was named JG3862 (*Table 1*). Genome sequencing of the deletion mutant strain did not reveal the significant introduction of additional mutations or DNA modifications (*Table 2*; *Figure 1b*). The WT and ΔSfPat strains were then used for in vitro and in vivo experiments to understand the potential role of prophages in shaping *S. fidelis* 3313 colonization dynamics in the gut of *Ciona*.

### Prophage deletion modulates biofilm formation and motility in *S. fidelis* 3313 in vitro

Deletion of SfPat from *S. fidelis* 3313 contributed to an overall increase in biofilm formation as quantified by crystal violet staining by over 14% compared to WT strains (*Figure 2a*). Conventionally, to form a biofilm, bacteria will settle and initiate stationary growth dynamics (*Watnick and Kolter, 2000*). We studied bacterial swimming on simple semi-solid media to determine if the prophage influenced swimming motility in *S. fidelis* 3313. Bacterial motility was measured by the spread diameter from a primary inoculation point after overnight incubation (*Figure 2b*). The WT strain demonstrated a mean diameter of 8.21 mm, while ΔSfPat resulted in a decrease in the mean diameter to be 4.04 mm, demonstrating reduced motility.

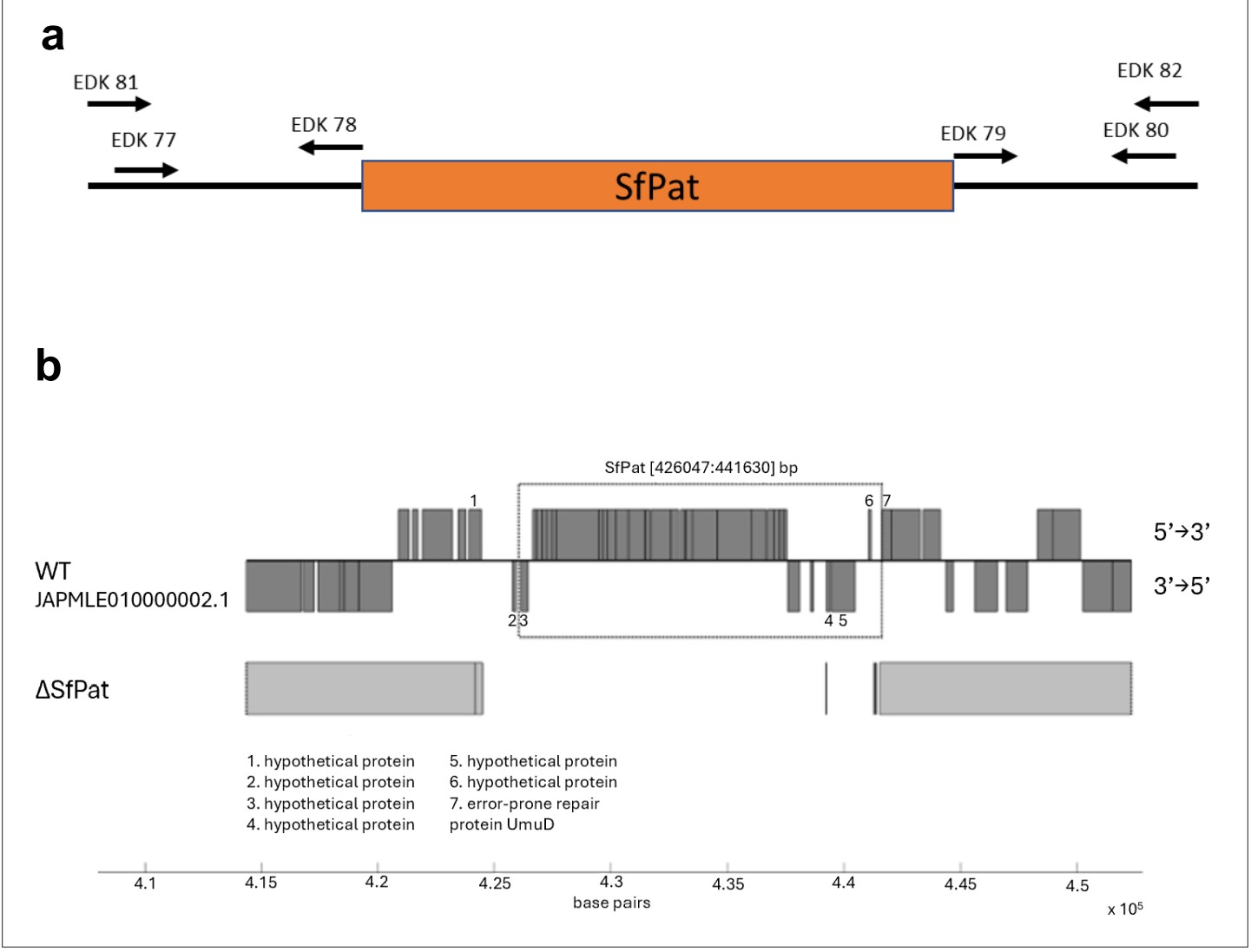

**Figure 1.** General prophage deletion scheme. (**a**) Location of upstream, downstream, and flanking primers used in the deletion of SfPat, primer orientation shown with respect to the prophage. (**b**) Deletion of SfPat from *S. fidelis* 3313 identified after assembling Illumina (short-read) sequenced genomes and mapping onto the improved (short and long-read, PacBio, sequencing) WT genome. The figure illustrates SfPat deletion as revealed by subsequent Illumina sequencing. The solid gray areas on the SfPat genome indicate regions that share identity with the WT.

## The influence of a prophage on colonization of the *Ciona* gut by *S. fidelis* 3313

Swimming and biofilm formation behaviors also depend on the modulation of cyclic-di-GMP in the bacteria. Thus, changes in the transcript levels of four genes regulating cyclic-di-GMP (*pleD*, *pilZ*, chitinase, and *pdeB*) were measured by qPCR between WT and ΔSfPat. We aimed to identify regulators of the transition from sessile to motile behavior impacted by the deletion of SfPat. No significant change in the transcript levels of the genes in bacteria recovered from 24-hour biofilms (in vitro)

**Table 1.** All *S. fidelis* 3313 strains are submitted under the BioProject PRJNA 90327 on NCBI, accession: SAMN31793880 ID:31793880.

| Organisms | Phenotype | NCBI RefSeq assembly |
|---|---|---|
| JG4066 | WT | GCF_033441085.1 |
| JG3862 | ΔSfPat | GCF_033441065.1 |

**Table 2.** Primer sequences on *S. flidelis* 3313 used for generating ΔSfPat deletion suicide vector and for deletion verification.

| Primer ID | Sequence (5′–3′) |
| --- | --- |
| EDK77 | AAATGGATCCCGATCAG CCTGCTAGTTTATT |
| EDK78 | ACGGAATAGGTTGAAT GCGACTCAGGC |
| EDK79 | TCGCATTCAACCTATTCCG TCATGTTTAGCC |
| EDK80 | ACATGAGCTCGATGCAGA TAAAGAGCCGTAAA |
| EDK81 | GTTTATTTTGTGGCAATCGCA |
| EDK82 | GGTAGCAGTGCTTAAACGAT |

(*Figure 2c*, *Figure 2—figure supplement 1s1a*) was identified. However, when these strain variants were introduced to metamorphic stage 4 (MS4) *Ciona* for 24 hours, the RT-qPCR revealed significant changes in the transcript levels of *pdeB* (*Figure 2c*, *Figure 2—figure supplement 1s1a*). The bacterial gene *pdeB* encodes a phosphodiesterase enzyme that degrades cyclic-di-GMP. By reducing cyclic-di-GMP, pdeB serves as a positive regulator of motility and a negative regulator of biofilm formation (*Chao et al., 2013*). The *pdeB* transcript levels were higher in the colonizing WT *S. fidelis* 3313 than the ΔSfPat mutant strain (*Figure 2c*).

Swimming and biofilm formation often facilitate bacterial colonization of a host. We investigated whether prophages could impact the ability of *S. fidelis* 3313 to colonize the *Ciona* gut. Colonization assays were performed on MS4 *Ciona* juveniles by exposing animals to WT or ΔSfPat strains, repeating the experiments six times to account for diverse genetic backgrounds (*Figure 3a*), that is, using gametes from distinct outbred adults.

After exposure to the bacterial strains, retention was estimated by recovering bacteria from animals and quantifying colony-forming units (cfus) at different time points. The ΔSfPat strain revealed a statistically insignificant 1.3-to-1.5-fold change in retention compared to the WT strain after 1 hour of bacterial exposure, a time point that mimics initial colonization (*Figure 3a*). However, 24 hours after exposure, WT was over twofold retained in the gut than the ΔSfPat strain (p<0.05) (*Figure 3a*).

To visualize the localization of WT and ΔSfPat mutant strains in the gut, juveniles of MS4 *Ciona* were exposed for 1 hour to BacLight Green-stained WT and BacLight Red-stained ΔSfPat strain variants and vice versa (*Figure 3b–d*). The 1-hour time point reflects changes in the initial colonization of juveniles. These experiments revealed a differential localization to the stomach epithelial folds by the WT and ΔSfPat mutant strains. The WT strain typically prefers to occupy the pyloric cecum and the posterior portion of the esophagus and entrance into the stomach (*Figure 3d and e*, *Figure 3—figure supplement 1s3a and s3b*). Retention of the ΔSfPat mutant was noted at the walls of the stomach during co-exposure (*Figure 3c*). Retention of the ΔSfPat mutant was observed in the stomach and intestines, and less in the esophagus (*Figure 3—figure supplement 2s4a and s4b*). These studies suggest spatial and temporal differences in retention by differentially lysogenized strains of *S. fidelis* 3313.

## Host immune discrimination and impact on lysogenized bacteria

Host immunity also plays an important role in shaping gut homeostasis. Distinct microbes and their antigens and/or metabolites can elicit host immune responses (*Rooks and Garrett, 2016*). To determine if the *Ciona* immune system discriminates among *S. fidelis* 3313 strains differing only in the presence or absence of the SfPat prophage, we examined the transcript levels of a secreted immune effector, VCBP-C, among juvenile MS4 during intestinal colonization. Under normal healthy conditions, VCBP-C is expressed and secreted by the gut epithelium and can bind (and opsonize) bacteria within the gut lumen (*Dishaw et al., 2011*) and influence biofilms in vitro (*Dishaw et al., 2016*). After 1 hour of exposure to the *S. fidelis* 3313 WT and mutant strains, changes were detected in the transcript levels of VCBP-C. Upregulation of VCBP-C was observed when juveniles were exposed to ΔSfPat mutant strains of *S. fidelis* 3313 compared to the WT strain (*Figure 4a*) by qPCR (p<0.05). As VCBP-C

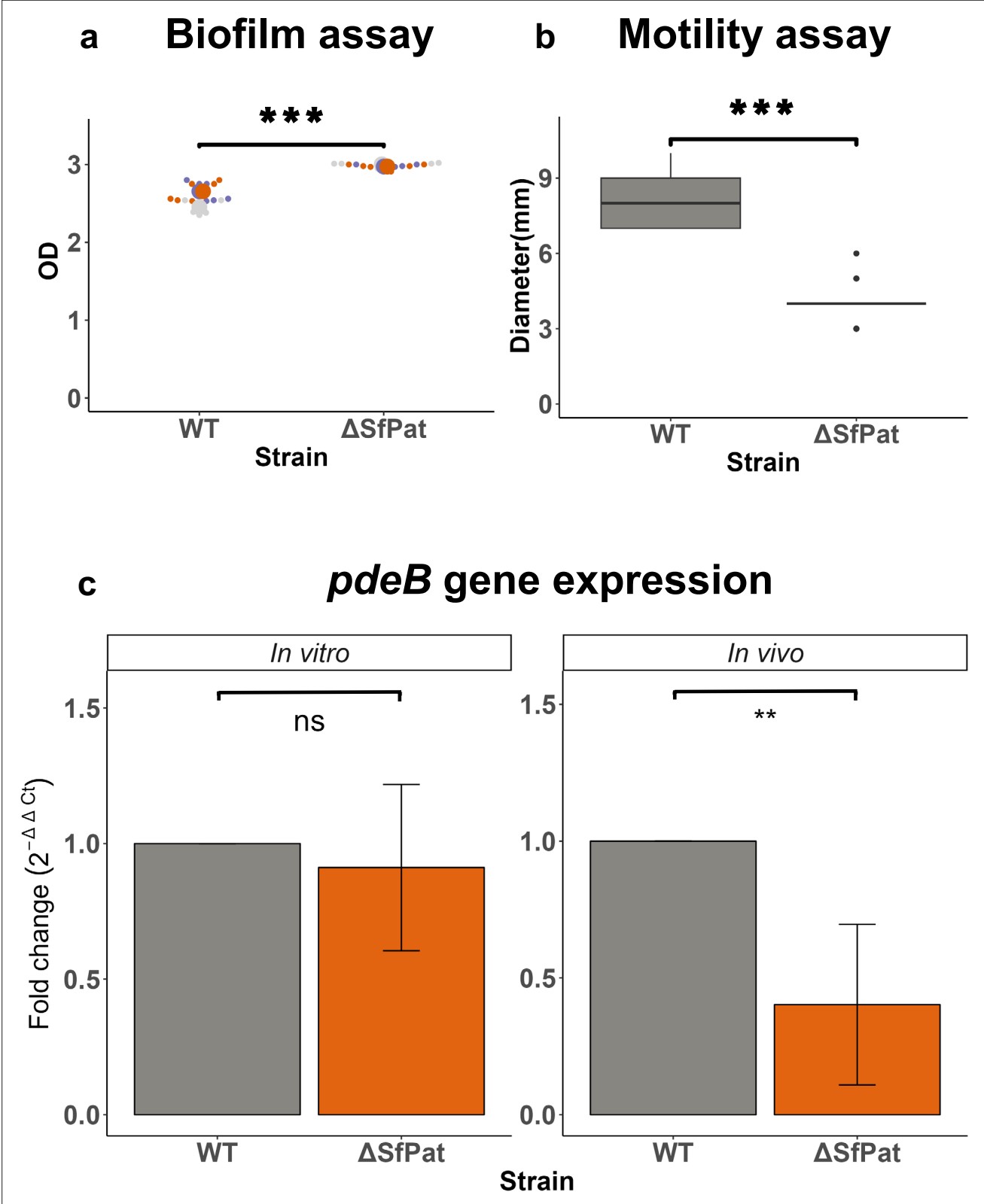

**Figure 2.** Effects of prophages on biofilm and swimming in *S. fidelis* 3313. (**a**) Influence of prophages on in vitro biofilm formation over 24 hours quantified with crystal violet assay (n=3), (**b**) role of prophages in swimming quantified as the diameter of spread on soft agar after 24 hours (n=6), and (**c**) fold-change of *pdeB* (with *Rho* as internal control) from 24 hour biofilm (in vitro) (n=4) and 24 hour in vivo (n=4). *p-value<0.05, **p-value<0.01; ns = not significant.

*Figure 2 continued on next page*

*Figure 2 continued*

The online version of this article includes the following figure supplement(s) for figure 2:

**Figure supplement 1.** qPCR showing relative fold change gene expression with Rho as an endogenous control.

is a major secreted effector of the gut regulating microbial settlement dynamics, the transcript levels of other innate immune genes were evaluated after 24 hours of exposure to the strains and did not reveal statistically significant responses. However, the lack of statistically significant responses may also be attributed to host genetic diversity, that is, differing responses to the same strains can obscure signal in transcript pools (*Figure 4b*).

Since the presence of SfPat influences host VCBP-C responses, we investigated whether the binding of VCBP-C to bacterial cell surfaces could influence prophage gene expression. *S. fidelis* 3313 was grown in MB in vitro for 24 hours in the presence or absence of 50 µg/ml of recombinant VCBP-C (*Dishaw et al., 2016*). The supernatant was discarded. RNA was extracted from the biofilms, and transcript levels among SfPat open reading frames were monitored, as well as the SOS response regulators, *lexA* and *recA*, in *S. fidelis* 3313. Adherent cultures, or biofilms, were analyzed rather than the supernatant since preliminary experiments revealed that the growth of WT *S. fidelis* in stationary cultures is influenced by VCBP-C. Interaction of VCBP-C with the WT strain was found to suppress the expression of the structural phage protein P5 of SfPat (*Figure 5*). It was noted that VCBP-C did not significantly alter the expression of *lexA* and *recA*, indicators of the SOS pathway (*Figure 5*). This suggests that VCBP-C binding to the surface of the *S. fidelis* 3313 strain may not influence prophage structural genes via conventional SOS responses.

## Discussion

In this report, we utilized a phage-deletion strategy to study the influence of a prophage (SfPat) on a gut symbiont (*S. fidelis* 3313) of the model invertebrate, *C. robusta*. Using RAST annotations and deduced amino acid BLAST (BLASTx), we propose that SfPat is a distant relative of the PM2 prophage (*Supplementary file 1*, *Kivelä et al., 2002*). Deletion of the PM2-like SfPat prophage from *S. fidelis* 3313 revealed phage-mediated influence on colonization dynamics. We find that deletion of the PM2-like prophage decreases swimming behaviors while increasing biofilm formation. These bacterial phenotypes influence host immune responses in ways that may influence differential retention in the gut of *Ciona*. Our identification of a prophage that interferes with biofilm formation in a *Shewanella* strain contrasts with reports demonstrating an increase in biofilms resulting from the incorporation of exogenous DNA following cell lysis (*Gödeke et al., 2011*; *Liu, 2019*). Our findings implicate prophage-induced motility changes in *Shewanella* spp. and add to the growing awareness that prophages can impact their hosts by modulating traits via diverse mechanisms. An influence on motility has also been linked to a prophage in *P. aeruginosa* (*Tsao et al., 2018*).

SfPat in *S. fidelis* 3313 appears to have a variable influence on bacterial retention in the gut of *Ciona*. As feeding is initiated in MS4 *Ciona* juveniles, food (or bacteria in the environment, natural or artificially introduced) accumulates in the gastrointestinal tract and on average takes about 45 minutes to begin exiting the anus and atrial siphon as fecal pellets. Thus, we monitored and compared transit and retention of introduced bacteria in the gut of MS4 juveniles at 1 and 24 hours after introduction. SfPat deletion reveals a prophage-mediated influence on gut retention and localization. For example, within 1 hour of exposure, fecal pellets begin to form in the stomach that are enriched for the WT *S. fidelis* 3313; however, the WT strain then appears to be retained in the posterior portion of the esophagus (just before the entrance into the stomach) as well as in the pyloric cecum (which is a small outpouching just ventral and posterior to the stomach). Instead, the ΔSfPat mutant strain appears to be retained in the stomach folds and the intestines, and not the esophagus. It should be noted that the base of the stomach folds is where secretion of VCBP-C occurs (*Dishaw et al., 2016*; *Dishaw et al., 2011*; *Liberti et al., 2015*). The overall retention of the two strains did not vary significantly within the first hour of colonization. However, after 24 hours, the retention of the WT strain was significantly greater than that of ΔSfPat despite the continued feeding behavior of the MS4 juveniles. The reduced transcript levels of *pdeB* in the SfPat deletion strain recovered from colonized animals might suggest that the secondary messenger, cyclic-di-GMP, is being differentially regulated in the

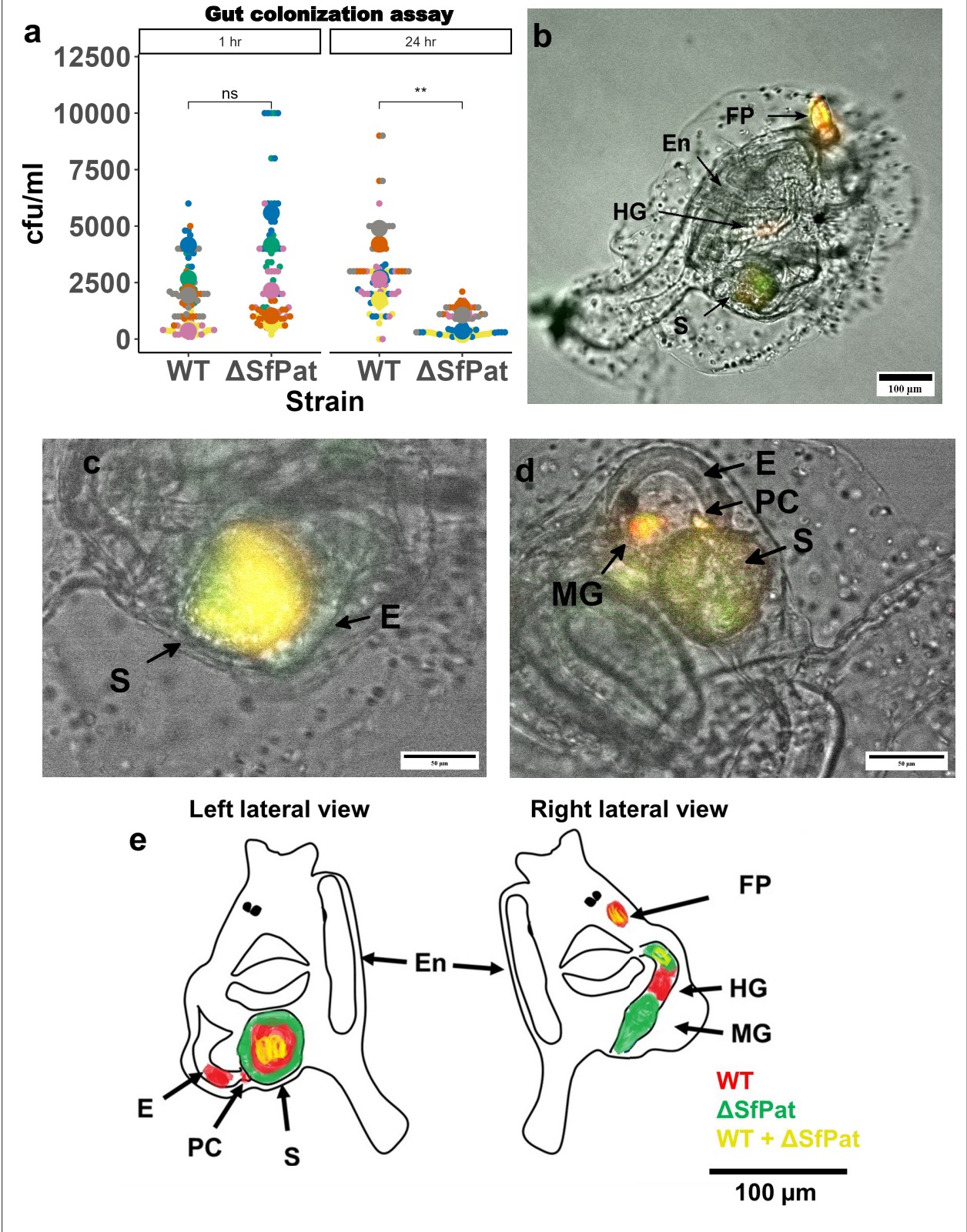

**Figure 3.** The influence of SfPat prophage on gut colonization in *Ciona*. (**a**) Results of six biological replicates (n=6, each replicate being a pool of ten juvenile tunicates) of the experimental exposure of *Ciona* MS4 juveniles to either WT or ΔSfPat strains for 1 hour and 24 hours; retention quantification displayed as a Beeswarm plot of colony-forming units (CFUs). There is significant retention observed in WT after 24 hours. The MS4 juveniles reveal differential colonization of WT and ΔSfPat after 1 hour of exposure (**b–e**), where WT strain is stained with BacLight Red and ΔSfpat is stained with

*Figure 3 continued on next page*

*Figure 3 continued*

BacLight Green reveal (**b**) WT is seen localized in the lower esophagus to anterior stomach, while the ΔSfPat deletion strain localized to the hindgut, while (**c**) the WT is seen localized mostly as a fecal pellet in the center of the stomach, while ΔSfPat prefers to localize to the stomach wall. (**d**) The WT strain is retained in the pyloric cecum. (**e**) Summary schematic of asymmetric bilateral views of MS4 animals; top of image is anterior and stomach is posterior. The ventral side is the 'En' side, and the dorsal side is the opposite side. The findings can be summarized as such: WT is retained in E and S, in PC, and also in the HG, while the ΔSfPat is retained in the stomach folds, MG, and portions of the HG. Some overlap in signal is noted with yellow coloring. En = endostyle, E = esophagus, S = stomach, MG = mid gut, HG = hind gut, PC = pyloric cecum.

The online version of this article includes the following figure supplement(s) for figure 3:

**Figure supplement 1.** BacLight-stained WT localization in *Ciona* MS4 after 1-hour exposure.

**Figure supplement 2.** BacLight-stained ΔSfPat localization in *Ciona* MS4 after 1-hour exposure.

two strains in longer-term retention and colonization assays. SfPat also appears to influence niche preference and retention patterns (*Figure 3*).

In addition to host genetics obscuring the influence of prophages on colonization of the gut, other biophysical factors that include host immune effectors play crucial and often silent roles in influencing bacterial settlement dynamics. For example, human secretory immunoglobulin A (SIgA) has been shown to enhance and often favor settlement of bacteria both in vitro and in vivo (*Bollinger et al., 2006*; *Thomas and Parker, 2010*; *Pratt and Kolter, 1998*; *Donaldson et al., 2018*), raising a basic question as to whether this phenomenon is more widespread among other secretory immune effectors present in mucosal environments of animals (*Dishaw et al., 2014a*). We speculate that while prophages likely impact the behavior of lysogenized bacteria in ways that can influence colonization dynamics, interaction with VCBP-C on the mucosal surface of the *Ciona* gut likely further influences settlement behaviors (*Dishaw et al., 2016*). Importantly, we show here that the influence of the SfPat prophage on bacterial physiology (i.e., the WT strains) leads to a reduced expression of *Ciona* VCBP-C in the first hours of bacterial retention in the gut (an indicator that the host has detected the exposure). It remains to be shown if prophages stimulate the production of a bacterial metabolite with immunomodulatory properties or if the host immune system responds to differences in bacterial behaviors or traits, as suggested in the ΔSfPat deletion mutant.

Metagenomic sequencing of gut microbes from healthy humans has revealed that temperate lifestyles are prevalent among phages from these ecosystems (*Minot et al., 2013*; *Minot et al., 2011*; *Reyes et al., 2010*), an observation also made in the *Ciona* gut (*Leigh et al., 2018*). Various environmental triggers, such as UV light and mutagenic agents like mitomycin C, have been shown to induce a switch from the temperate to lytic cycle via the SOS response, a cell-wide response to DNA damage that can promote survival (*Weinbauer and Suttle, 1999*). Since VCBP-C is an immune molecule in the gut that can interact with bacteria, it could influence prophage induction. However, we find that VCBP-C binding on the surface of WT *S. fidelis* 3313 leads to a reduction in the expression of an important SfPat structural protein P5, suggesting a limitation in SfPat induction in the presence of VCBP-C. No significant changes in *lexA/recA* expression were observed upon VCBP-C exposure/binding, suggesting a lack of SOS response when exposed to this immune effector. The various mechanisms by which prophages shape colonization behaviors among gut bacteria of animals remain unclear. While the data reported here are only based on one bacterial strain that colonizes the *Ciona* gut, we find that WT *S. fidelis* colonizes the gut with reduced activation of VCBP-C gene expression compared to ΔSfPat, a trait that may be important in shaping colonization outcomes. We speculate that these observations are more widely applicable since lysogens are so abundant in animal microbiomes. Under normal conditions, VCBP-C protein is present in copious amounts and tethered to chitin-rich mucus lining the gut, as revealed by immunohistochemical staining (*Dishaw et al., 2016*). Therefore, overexpression of VCBP-C is not necessarily helpful and can correspond to the induction of additional inflammatory responses, including an overproduction of mucus. Thus, regulation of the production of additional VCBP-C likely serves important roles in influencing colonization dynamics.

Since colonization of animal mucosal surfaces is an ancient process (*Dishaw et al., 2014a*; *Dishaw et al., 2014b*), prophages and their integration into bacterial genomes have likely evolved to provide fitness benefits in often challenging environments like the gut lumen. Determining the role of animal immunity and prophages in these exchanges is of broad interest. Immune effectors like VCBPs, which undoubtedly possess broad specificities, can bind a range of bacterial hosts; however, it remains to be shown if they bind lysogenized bacteria with different affinities than their prophage-free counterparts.

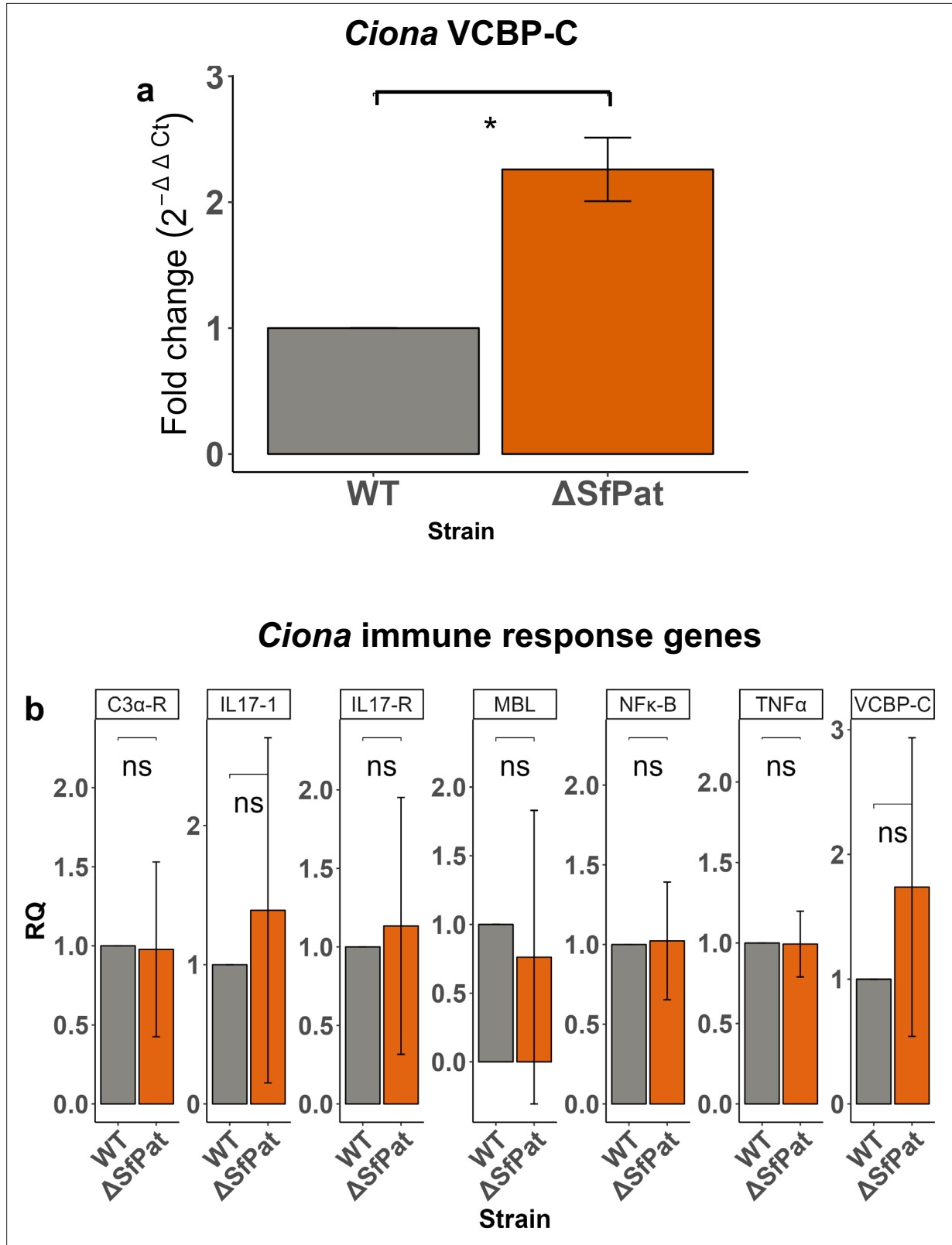

**Figure 4.** The influence of prophages on host gene expression. (**a**) VCBP-C gene expression in MS4 juveniles after 1 hour of exposure to *S. fidelis* 3313 strains (n=4). (**b**) Survey of additional innate immune gene expression in MS4 juveniles after 24-hour exposure to WT or ΔSfPat mutant strains (n=3). Actin is the internal control. *p-value<0.05, **p-value<0.01, ns = not significant.

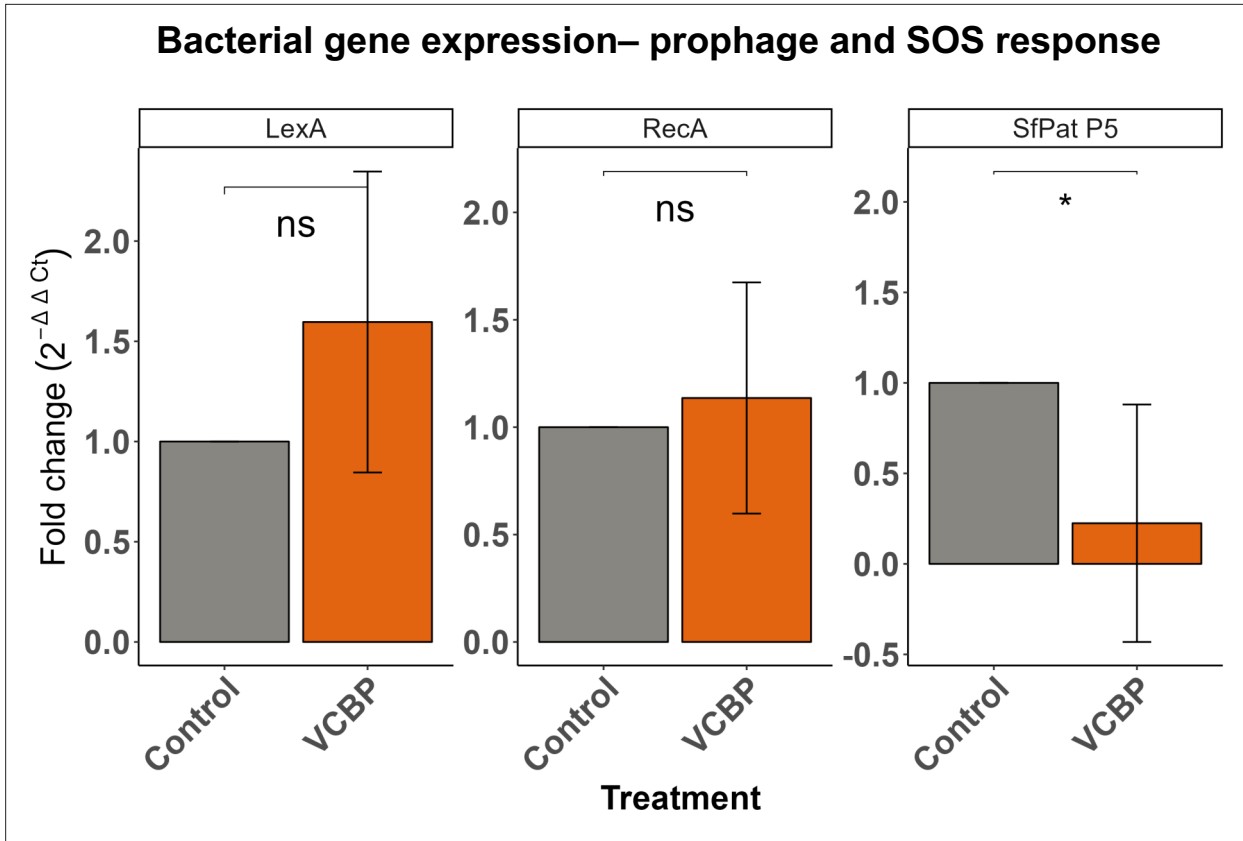

**Figure 5.** Lysogen gene expression in response to host immune effector binding. Gene expression of SfPat structural protein P5, *recA* and *lexA* of WT strain grown as a 24-hour biofilm while exposed to 50 µg/ml VCBP-C. Rho is the internal control (n=4). *p-value<0.05, ns = not significant.

The online version of this article includes the following figure supplement(s) for figure 5:

**Figure supplement 1.** RefFinder identifying Rho as the stable endogenous control.

Prophages can also be induced to generate lytic particles that can influence gut microbiome structure, serving as an indirect form of protection for the host (*Wang et al., 2010*; *Barr et al., 2013*). Prophages can also contribute to the transfer of virulence factors (*Nanda et al., 2015*; *Wagner and Waldor, 2002*). Retention of lysogens may be preferred if the prophages provide competitive fitness and retention in the gut. Since lysogens are integral in animal development, immunity, and metabolism (*Fraune and Bosch, 2010*), there is a tripartite interplay required for survival, a snapshot of which is shown here.

## Materials and methods
### Culture and growth conditions

*S. fidelis* 3313 used in this study was originally isolated from the gut of *C. robusta* obtained from Mission Bay, CA, USA, as previously described (*Leigh et al., 2017*). The bacterium was cultured using Difco marine agar 2216 (MA) (Fisher Scientific, Hampton, NH) and marine broth (MB) at room temperature (RT) (22–24°C). Subsequent genetic manipulations were performed on strains grown in

**Table 3.** Plasmids and strains used in the study.

| Plasmids/ strains | Genotype | Source/reference |
|---|---|---|
| *E. coli* UQ950 | *E. coli* DH5α λ (pir) host for cloning; FΔ(argFac)169 Φ80dlacZ58(ΔM15) glnV44(AS)rfbD1 gyrA96(NalR) recA1endA1spoT1 thi1 hsdR17 deoR Λ pir+ | *Saltikov and Newman, 2003* |
| pSMV3⁻ᐞ SfPat | pSMV3 with 778 bp upstream and 779 bp downstream of flanking regions of SfPat | This study |

LB/MB, which consists of a mixture of 75% LB (Lysogeny Broth [Luria], Fisher Scientific, Hampton, NH) and 25% MB. Strains are listed in *Table 1*.

## Prophage deletion

SfPat was targeted for deletion from *S. fidelis* 3313 using homologous recombination methods adapted from *Saltikov and Newman, 2003* to produce knockout mutant strains. First, a pSMV3 suicide vector (*Saltikov and Newman, 2003*) was designed with ≈700 bp regions corresponding to the upstream and downstream sequence of the prophage (*Table 2*). These flanking regions were amplified and ligated using overlap extension PCR, then directionally inserted into the vector with the restriction enzymes BamHI and SacI (*Table 3*; *Bryksin and Matsumura, 2010*). Plasmid conjugation was then performed by inoculating a colony of *S. fidelis* 3313 into a culture of *E. coli* containing the desired suicide vector on an LB/MB agar plate for 2 hours followed by primary selection after 24 hours at RT on LB/MB +100 µg/ml kanamycin plates and a counter selection on LM/MB +10% Sucrose at RT. Merodiploids and deletion mutants were verified by PCR. Illumina sequencing (MiGS, University of Pittsburgh) confirmed the deletion of SfPat and the evaluation of any additional genetic changes or mutations (*Figure 1b*).

## *S. fidelis* crystal violet biofilm assay

WT and ∆SfPat strains were cultured in MB overnight at RT and then diluted to $10^7$ cfu/ml in MB. Cultures were brought to a final volume of 2 ml of MB in 12-well dishes and incubated at RT for 24 hours to examine biofilm development. Each variable was tested in technical duplicate. The biofilms were quantified by crystal violet staining as previously described (*Liberti et al., 2022*). Briefly, the supernatants were aspirated after 24 hours of incubation, and the biofilms were dried and stained with 0.1% crystal violet for 10 minutes. The stained biofilms were then gently washed with deionized water, and the amount of biofilm produced was quantified as the intensity of the stain ($OD_{570}$) after the biofilm-bound crystal violet was extracted from the biofilm with 30% acetic acid. All biofilm assays were performed at least in triplicates.

## Motility assay

Soft-agar overlay motility assays were carried out in 12-well dishes to compare swimming behaviors (*Wolfe and Berg, 1989*; *Kearns, 2010*). Briefly, a single colony from an overnight streaked LB/MB plate was picked using a sterile toothpick and then stabbed onto the center of soft agar (containing LB/MB and 0.5% low-melt agarose) and incubated at RT overnight. The results were recorded as the distance traveled (in millimeters) by the bacteria from the inoculation zone. Each variable was tested in duplicate. Two perpendicular distances from the inoculation zone were recorded for each technical replicate and averaged for each well.

## *Ciona* mariculture

The in vivo colonization experiments were performed on animals reared under conditions termed 'semi-germ-free' (SGF), which include minimal exposure to marine microbes. SGF conditions include animals harvested under conventional approaches (*Cirino et al., 2002*) but permanently maintained in 0.22 µm-filtered, conditioned artificial seawater (cASW), handled with gloves, and lids only carefully removed for water/media changes. cASW is prepared by conditioning ASW made with Instant Ocean in an in-house sump-aquarium system containing live rock, growth lights, and sediment from San Diego, California; salinity is maintained at 32–34 parts per thousand (or grams per liter). Compared to germ-free (*Leigh et al., 2016*) or SGF, conventionally-reared (CR) includes a step-up exposure to 0.7 µm-filtered cASW that increases exposure to marine bacteria during development. The SGF approach is considered an intermediate method of rearing that includes minimal exposures to microbial signals during development (unpublished observations). The animals were reared at 20°C from larval to juvenile stages. The *Ciona* were collected from Mission Bay, CA, in order to produce juvenile organisms for each biological replicate. These wild-harvested animals provide a wider genetic diversity compared to traditional model systems, where genetic diversity has been reduced or eliminated through controlled breeding practices.

## Gut colonization assays

Both bacterial strains were grown overnight at RT in MB and diluted to $10^7$ cfu/ml in cASW after repeated washes. Metamorphic stage 4 (MS4) animals reared in six-well dishes in cASW were exposed

to 5 ml of $10^7$ cfu/ml bacteria in each well for 1 hour or 24 hours. Co-exposure studies were also performed, where 2.5 ml (or 1:1) of each culture prepared above was mixed to form a total volume of 5 ml. MS4 animals are considered part of the first ascidian stages (post-settlement stages 1–6, whereas stages 7–8 and onward are second ascidian stages and reflect young adult animals). MS4 juveniles can be identified as having a pair of protostigmata, or gill slits, on each side of the animal (*Chiba et al., 2004*). These juveniles first initiate feeding via newly developed and opened siphons; before this, the gut remains closed, and the interior lumen is unexposed to the outside world. Following this initial exposure or colonization, for various time intervals, the plates were rinsed multiple times with cASW and replaced with fresh cASW. Ten juveniles were chosen randomly for each treatment, pooled, and homogenized with a plastic pestle; live bacteria were counted by performing serial dilutions and enumerating cfus via spot-plating assays (*Gaudy et al., 1963*; *Miles et al., 1938*). Each graphed data point represents a biological replicate dataset from genetically distinct/diverse backgrounds of *Ciona* (represented by separate live animal collection and spawning events). Statistical significance was calculated using the Wilcoxon *t*-test by pooling data across six genetically diverse biological replicates.

Live bacteria in the gut were visualized using BacLight stains and previously described fluorescently labeled bacteria (*Moran et al., 2019*). For BacLight staining, 1 ml of bacterial cultures was grown overnight at RT, pelleted, washed twice with cASW, and stained with 4 µl of BacLight Red (Invitrogen, Cat# B35001) or BacLight Green (Invitrogen, Cat# B35000) for 15 minutes in the dark. The cultures were stained with alternate dyes in different replicates to get unbiased data from changes in fluorescence. The stained cultures were washed twice with cASW, and then diluted to $10^7$ cfu/ml with cASW. MS4 animals grown in six-well dishes were then exposed to 5 ml of this culture. Bacteria in the gut of animals were visualized after 1 hour on a Leica DMI 6000B stereoscope with a CY5 fluorescent filter for BacLight Red and GFP filter for BacLight Green; and imaged and captured with a Hamamatsu ORCAII camera (model C10600-10B-H) and processed with the MetaMorph 7.10.4 imaging suite (Molecular Devices, Downingtown, PA).

## Differential transcript-level studies

To determine if the *Ciona* innate immune system can recognize and respond to unique mutant strains, which differ only in the presence or absence of prophages, candidate immune response

**Table 4.** *Ciona* genes targeted and the necessary reverse transcription-qPCR primers.

| Gene | Function | GenBank accession no. | Primer (5'–3') | Reference |
|---|---|---|---|---|
| VCBP-C-Fwd | Secreted immune effector in the gut | HQ324151 | AGACCAACGCCAACACAGTA | *Liberti et al., 2018* |
| VCBP-C-Rev | | | CCCCATACATTGCAGCATTTC | |
| Actin–Fwd | Cytoskeletal actin | AJ297725 | CCCAAATCATGTTCGAAACC | *Liberti et al., 2018* |
| Actin-Rev | Reference gene | | ACACCATCACCACTGTCGAA | |
| IL17-1-Fwd | Interleukin 17 | NM_001129875.1 | AGGTTAAGAATCCCTATGGTGC | *Liberti et al., 2023* |
| IL17-1-Rev | Effector cytokine | | CAAAGGCACAGACGCAAAGG | |
| IL17-1R-Fwd | Interleukin 17 receptor | NM_001245045 | TGTTGGCATGAGTGTTCGGT | |
| IL17-1R-Rev | | | AGTTGGTTCTGCCCCAAAGT | |
| NFκ-B-Fwd | Immune regulatory | NM_001078304 | TGTCGCTTGTCGTCATGGAA | *Liberti et al., 2023* |
| NFκ-B-Rev | Transcription factor | | AACACCCAAGACCGTCGAAA | |
| TNFα-Fwd | Tumor necrosis factor | NM_001128107 | TTCAGAAAGATTGGACGACGA | *Liberti et al., 2023* |
| TNFα-Rev | Inflammatory effector | | TCGTTTAGAAATGCTGCTGTGG | |
| C3A-R-Fwd | Complement C3 | NM_001078552 | TTGTAAGCTGGCACAAGGTGT | This study |
| C3A-R-Rev | Inflammatory mediator | | GACCGTAGTCTGGTAGAGGTC | |
| MBL-Fwd | Mannose binding lectin | NM_001167707.2 | TTATTGATGGGAAAGTTTGGT | This study |
| MBL-Rev | | | TAACATCTCTGTTCTTGGGTC | |

**Table 5.** *S. fidelis* 3313 genes targeted and the necessary reverse transcription-qPCR primers.

| Gene | Function | Genbank accession no. | Primer (5'–3') | Reference |
|------|----------|----------------------|----------------|-----------|
| pleD-Fwd | Regulates production of cyclic di-GMP | NZ_KI912459.1 | CTCTTCACCGCCACTTCTT | This study |
| pleD-Rev | | | GGTGTGGTCTCTTATGCCTATC | |
| Chitinase-Fwd | Chitin utilization gene | NF027718.3 | CAGTGTAGCTAAGTCGTCATC | This study |
| Chitinase-Rev | | | CGCCAACCAGTGCTTTATTG | |
| pilZ-Fwd | Type IV Pilus control protein | NZ_JADX01000014.1 | TGGCAAGGTCGTTTGGATTA | This study |
| pilZ-Rev | | | AGGCAAGCTCACTGGAAAG | |
| pdeB-Fwd | Phosphodiesterase | NF012772.3 | GCATCAGGGCTCTTACCAATAG | This study |
| pdeB-Rev | | | GAGGCGGTGATCCTTACAGATA | |
| RecA-Fwd | Bacterial recombinase/bacterial reference gene | NZ_KI912459.1 | CGTAGTGGTGCGGTAGATGT | This study |
| RecA-Rev | | | CGCATTGCTTGGCTCATCAT | |
| LexA-fwd | Regulator of recombinase | NZ_JADX01000011.1 | TGACCCAGCTATGTTCCGCC | This study |
| LexA-Rev | | | GCTCAACCTTGTGTACGGCG | |
| Rho-Fwd | Bacterial reference gene | NZ_JADX01000026.1 | CACGTACAAGTTGCCGAAATG | This study |
| Rho-Rev | | | CAAGACGGGTGATAGAGTCAAG | |
| gyrB-Fwd | Bacterial reference gene | AM229309.2 | TCACGAGCATCATCACCCGT | This study |
| gyrB-Rev | | | GGCTTCCGTGGTGCGTTAAC | |

markers were examined using reverse transcription quantitative PCR (RT-qPCR). RNA was extracted using the RNA XS kit (Macherey-Nagel, Cat# 740902) from MS4 *Ciona* juveniles exposed to either the WT or ΔSfPat strain. Complementary DNA (cDNA) synthesis was performed with oligo-dT primers and random hexamers using the First Strand cDNA Synthesis Kit (Promega, Cat# A5000) following the manufacturer's instructions. The amplification was performed with the qPCR kit (Promega, Cat# A6000) and carried out on an ABI7500 with an initial melting temperature of 95°C for 2 minutes and 40 cycles of 95°C for 15 seconds and 60°C for 1 minute. The innate immune genes examined and their primers are reported in *Table 4*. Results from four distinct biological replicates are presented. Each replicate includes pooled *Ciona* juveniles from at least two wells of a six-well dish. *Ciona* actin was referenced as an endogenous control. Data was analyzed using the ΔΔCt method (*Pfaffl, 2001*) and the ABI7500 software suite. For the ΔΔCt method, the Ct values were first normalized to an endogenous control gene and further normalized to the reference samples, here the WT strain.

To understand bacterial genes that are differentially regulated due to the presence or absence of prophages, transcript levels were studied in vitro and in vivo. The bacterial strains were grown in six-well dishes using the same methodology described for biofilm assays. To understand if the host immune molecule VCBP-C induced prophages, WT was cultured in six-well dishes as described in the biofilm section in the presence or absence of 50 µg/ml VCBP-C in Marine Broth (*Dishaw et al., 2016*). After 24 hours, the supernatant was discarded for both experiments, and RNA from the biofilm was extracted using the Zymo Research Direct-zol RNA mini prep kit (Cat# R2050). cDNA synthesis was performed using random hexamers as primers, and qPCR was conducted as described above. The targeted bacterial genes are described in *Table 5*. Different housekeeping genes were examined for the different treatments, and the Ct values were used to identify the most stable reference gene to be used as an endogenous control. Rho was identified as the most stable reference gene using RefFinder, which utilizes Bestkeeper, GeNorm, Normfinder, and comparative ΔΔCt methods (*Figure 5—figure supplement 1*; *Andersen et al., 2004*; *Pfaffl, 2001*; *Vandesompele et al., 2002*; *Watnick and Kolter, 2000*).

## Statistical analysis and data visualization

Statistical analysis and data visualization were carried out in R, version 4.2 (*R Development Core Team, 2021*). Data were plotted with ggplot 3.3.5 *Kassambara, 2020*; the Beeswarm plot was constructed using ggbeeswarm 0.6 (*Aron and Trimble, 2021*). Beeswarm plots and statistics for motility assays were calculated using replicate averages (*Lord et al., 2020*). Statistical significance was calculated using ggsignif package 0.6.3 or ggpubr0.4.0 (*Kassambara, 2020*; *Constantin I Ahlmann-Eltze and Patil, 2021*). If the data were found to be normally distributed by Shapiro's test, then significance was calculated using an unpaired *t*-test. The Wilcoxon signed-rank test was used to calculate the significance of non-parametric data.

## Acknowledgements

The authors acknowledge the expertise of Gary W Litman and John P Cannon for expert feedback and guidance on earlier efforts of the project, and anonymous reviewers who helped improve earlier versions of the manuscript.

## Additional information

### Funding

| Funder | Grant reference number | Author |
|---|---|---|
| National Science Foundation | IOS-1456301 | Mya Breitbart Larry J Dishaw |
| National Science Foundation | MCB-1817308 | Larry J Dishaw |
| National Science Foundation | NSF IOS-2226050/51 | Jeffrey A Gralnick Larry J Dishaw |
| National Science Foundation | Graduate Research Fellowship Program | Brittany A Leigh |

The funders had no role in study design, data collection and interpretation, or the decision to submit the work for publication.

### Author contributions

Ojas Natarajan, Conceptualization, Data curation, Formal analysis, Validation, Investigation, Visualization, Methodology, Writing – original draft, Project administration, Writing – review and editing; Susanne L Gibboney, Formal analysis, Investigation, Methodology, Project administration; Morgan N Young, Software, Formal analysis, Visualization; Shen Jean Lim, Formal analysis, Writing – review and editing; Felicia Nguyen, Natalia Pluta, Celine GF Atkinson, Eric D Kees, Investigation; Assunta Liberti, Validation, Methodology, Writing – review and editing; Brittany A Leigh, Conceptualization, Funding acquisition; Mya Breitbart, Conceptualization, Resources, Funding acquisition, Writing – review and editing; Jeffrey A Gralnick, Conceptualization, Resources, Funding acquisition, Methodology, Writing – review and editing; Larry J Dishaw, Conceptualization, Data curation, Supervision, Funding acquisition, Validation, Investigation, Visualization, Methodology, Writing – original draft, Project administration, Writing – review and editing

### Author ORCIDs

Ojas Natarajan ⬤ https://orcid.org/0000-0001-7887-2640
Jeffrey A Gralnick ⬤ https://orcid.org/0000-0001-9250-7770
Larry J Dishaw ⬤ https://orcid.org/0000-0002-2705-4573

### Ethics

Our research focused on C. robusta, a marine invertebrate collected in Mission Bay near San Diego, CA (M-REP, Carlsbad, CA, USA), in areas that are not privately owned or protected. Since Ciona is considered an invasive species, it is not protected by US environmental agencies. The collection service, M-Rep, maintains the necessary permits and licenses for collecting and distributing marine

invertebrates to academic institutions. Live animals were handled according to animal use guidelines (Policy 0-308) at the University of South Florida. The animals were collected and transported alive, then kept in clean water with aeration. As per standard animal protocols, the minimum number of animals necessary for each experiment was utilized. Animal waste products were disposed of following USF Health and Safety Guidelines.

Reviewer #1 (Public review): https://doi.org/10.7554/eLife.103107.3.sa1
Reviewer #2 (Public review): https://doi.org/10.7554/eLife.103107.3.sa2
Reviewer #3 (Public review): https://doi.org/10.7554/eLife.103107.3.sa3
Author response https://doi.org/10.7554/eLife.103107.3.sa4

# Additional files

## Supplementary files

Supplementary file 1. RAST annotations of the prophage SfPat.

MDAR checklist

## Data availability

All *S. fidelis* 3313 strains are submitted under the BioProject PRJNA90327 on NCBI, accession: SAMN31793880 ID:31793880 JG4066 - WT - GCF_033441085.1 (NCBI RefSeq assembly) JG3862-ΔS-fPat - GCF_033441065.1 (NCBI RefSeq assembly). Bacterial strains used in this study will be made available upon request under a Material Transfer Agreement (MTA) with the University of South Florida. All primers are listed in the published tables. We are also happy to share any updates to experimental protocols upon reasonable request.

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
