## [Editor Report · eLife Assessment]

This **valuable** study presents findings linking prophage carriage to lifestyle regulation in the marine bacterium *Shewanella fidelis*, with potential implications for niche occupation within a host (*Ciona robusta*) and mediation of host immune responses. The study leverages a unique animal model system that offers distinct advantages in identifying select phenotypes to present overall **solid** evidence that supports findings relating to the impact of a prophage on host-microbe interaction. Understanding the role of integrated lysogenic phages in bacterial fitness, both within a host and in the environment, is a significant concept in bacterial eco-physiology, potentially contributing to the success of certain strains.

---

## [Referee Report · Reviewer #1 (Public review)]

Summary:

The manuscript aims to elucidate the impact of a prophage within the genome of Shewanella fidelis on its interaction with the marine tunicate Ciona robusta. The authors made a deletion mutant of S. fidelis that lacks one of its two prophages. This mutant exhibited an enhanced biofilm phenotype, as assessed through crystal violet staining, and showed reduced motility. The authors examined the effect of prophage deletion on several genes that could modulate cyclic-diGMP levels. While no significant changes were observed under in vitro conditions, the gene for one protein potentially involved in cyclic-diGMP hydrolysis was overexpressed during microbe-host interactions. The mutant was retained more effectively within a one-hour timeframe, whereas the wild-type (WT) strain became more abundant after 24 hours. Fluorescence microscopy was used to visualize the localization patterns of the two strains, which appeared to differ. Additionally, a significant difference in the expression of one immune protein was noted after one hour, but this difference was not evident after 23 hours. An effect of VCBC-C addition on the expression of one prophage gene was also observed.

Strengths:

I appreciate how the authors integrate diverse expertise and methods to address questions regarding the impact of prophages on gut microbiome-host interactions. The chosen model system is appropriate, as it allows for high-throughput experimentation and the application of simple imaging techniques.

Weaknesses:

My primary concern is that the manuscript primarily describes observations without providing insight into the molecular mechanisms underlying the observed differences. It is particularly unclear how the presence of the prophage leads to the phenotypic changes related to bacterial physiology and host-microbe interactions. Which specific prophage genes are critical, or is the insertion at a specific site in the bacterial genome the key factor? While significant effects on bacterial physiology are reported under in vitro conditions, there is no clear attribution to particular enzymes or proteins. In contrast, when the system is expanded to include the tunicate, differences in the expression of a cyclic-diGMP hydrolase become apparent. Why do we not observe such differences under in vitro conditions, despite noting variations in biofilm formation and motility? Furthermore, given that the bacterial strain possesses two prophages, I am curious as to why the authors chose to target only one and not both.

Regarding the microbe-host interaction, it is not clear why the increased retention ability of the prophage deletion strain did not lead to greater cell retention after 24 hours, especially since no differences in the immune response were observed at that time point.

Concerning the methodological approach, I am puzzled as to why the authors opted for qPCR instead of transcriptomics or proteomics. The latter approaches could have provided a broader understanding of the prophage's impact on both the microbe and the host.

Comments on revisions:

While the authors were able to solve some of my issues, I see that other questions were not tackled.

---

## [Referee Report · Reviewer #2 (Public review)]

Summary:

In the manuscript, "Prophage regulation of Shewanella fidelis 3313 motility and biofilm formation: implications for gut colonization dynamics in Ciona robusta", the authors are experimentally investigating the idea that integrated viruses (prophages) within a bacterial colonizer of the host Ciona robusta affect both the colonizer and the host. They found a prophage within the Ciona robusta colonizing bacterium Shewanella fidelis 3313, which affected both the bacteria and host. This prophage does so by regulating the phosphodiesterase gene pdeB in the bacterium when the bacterium has colonized the host. The prophage also regulates the activity of the host immune gene VCBP-C during early bacterial colonization. Prophage effects on both these genes affect the precise localization of the colonizing bacterium, motility of the bacterium, and bacterial biofilm formation on the host. Interestingly, VCBP-C expression also suppressed a prophage structural protein, creating a tripartite feedback loop in this symbiosis. This is exciting research that adds to the emerging body of evidence that prophages can have beneficial effects not only on their host bacteria but also on how that bacteria interacts in its environment. This study establishes the evolutionary conservation of this concept with intriguing implications of prophage effects on tripartite interactions.

Strengths:

This research effectively shows that a prophage within a bacterium colonizing a model ascidian affects both the bacterium and the host in vivo. These data establish the prophage effects on bacterial activity and expand these effects to the natural interactions within the host animal. The effects of the prophage through deletion on a suite of host genes are a strength, as shown by striking microscopy.

Weaknesses:

Unfortunately, global transcriptomics of the bacteria and the host during colonization by the prophage-containing and prophage-deleted bacteria (1 hour and 24 hours) would be suggested to better understand the tripartite interactions.

Impact:

The authors are correct to speculate that this research can have a significant impact on many animal microbiome studies, since bacterial lysogens are prevalent in most microbiomes. Screening for prophages, determining whether they are active, and "curing" the host bacteria of active prophages are effective tools for understanding the effects these mobile elements have on microbiomes. There are many potential effects of these elements in vivo, both positive and negative, this research is a good example of why this research should be explored.

Context:

The research area of prophage effects on host bacteria in vitro has been studied for decades, while these interactions in combination with animal hosts in vivo have been recent. The significance of this research shows that there could be divergent effects based on whether the study is conducted in vitro or in vivo. The in vivo results were striking. This is particularly so with the microscopy images. The benefit of using Ciona is that it has a translucent body which allows for following microbial localization. This is in contrast to mammalian studies where following microbial localization would either be difficult or near impossible.

Comments on revisions:

I am satisfied with the great amount of work that went into the comments provided by the reviewers. The figure presentations are more compelling for the story, and this latest revision is a very interesting read that should be considered for future microbiome studies.

---

## [Referee Report · Reviewer #3 (Public review)]

In this manuscript, Natarajan and colleagues report on the role of a prophage, termed SfPat, in the regulation of motility and biofilm formation by the marine bacterium Shewanella fidelis. The authors investigate the in vivo relevance of prophage carriage by studying the gut occupation patterns of Shewanella fidelis wild-type and an isogenic SfPat- mutant derivative in a model organism, juveniles of the marine tunicate Ciona robusta. The role of bacterial prophages in regulating bacterial lifestyle adaptation and niche occupation is a relatively underexplored field, and efforts in this direction are appreciated.

Comments on revisions:

The authors have addressed my main concerns. While some responses remain somewhat ambiguous or defer key clarifications to future studies, I appreciate that not everything can be resolved within a single manuscript.

---

## [Author Response]

The following is the authors’ response to the original reviews.

**Reviewer #1 (Public review):**
Summary:The manuscript aims to elucidate the impact of a prophage within the genome of Shewanella fidelis on its interaction with the marine tunicate Ciona robusta. The authors made a deletion mutant of S. fidelis that lacks one of its two prophages. This mutant exhibited an enhanced biofilm phenotype, as assessed through crystal violet staining, and showed reduced motility. The authors examined the effect of prophage deletion on several genes that could modulate cyclic-diGMP levels. While no significant changes were observed under in vitro conditions, the gene for one protein potentially involved in cyclic-diGMP hydrolysis was overexpressed during microbe-host interactions. The mutant was retained more effectively within a one-hour timeframe, whereas the wild-type (WT) strain became more abundant after 24 hours. Fluorescence microscopy was used to visualize the localization patterns of the two strains, which appeared to differ. Additionally, a significant difference in the expression of one immune protein was noted after one hour, but this difference was not evident after 23 hours. An effect of VCBC-C addition on the expression of one prophage gene was also observed.Strengths:I appreciate how the authors integrate diverse expertise and methods to address questions regarding the impact of prophages on gut microbiome-host interactions. The chosen model system is appropriate, as it allows for high-throughput experimentation and the application of simple imaging techniques.Weaknesses:My primary concern is that the manuscript primarily describes observations without providing insight into the molecular mechanisms underlying the observed differences. It is particularly unclear how the presence of the prophage leads to the phenotypic changes related to bacterial physiology and host-microbe interactions.

We appreciate the overall, enthusiastic reviewer feedback. The current manuscript presents experimental evidence of the biological impact of the deletion of a stably integrated prophage in the genome of Shewanella fidelis 3313. The molecular mechanisms responsible for these biological effects are currently unknown but based on the limited genetic insight of some predicted gene regions, we can speculate on prophage-mediated influences impacting swimming behaviors. Below, we address additional concerns raised by the reviewer.

Which specific prophage genes are critical, or is the insertion at a specific site in the bacterial genome the key factor? While significant effects on bacterial physiology are reported under in vitro conditions, there is no clear attribution to particular enzymes or proteins.

In this particular case, it is not entirely clear, as most ORFs within the prophage region have unknown functions, i.e., predicted as hypothetical proteins. In addition, the original insertion site does not appear to interrupt any specific gene but may impact adjacent genes/pathways (Fig 1b). Enhanced annotations, along with future targeted deletion methods for distinct prophage segments, will help us better investigate which predicted gene regions influence the observed traits. This will deepen our understanding of the mechanisms that regulate prophage influence on these traits.

In contrast, when the system is expanded to include the tunicate, differences in the expression of a cyclic-diGMP hydrolase become apparent. Why do we not observe such differences under in vitro conditions, despite noting variations in biofilm formation and motility? Furthermore, given that the bacterial strain possesses two prophages, I am curious as to why the authors chose to target only one and not both.

Differences in expression patterns of c-di-GMP regulators were also noted in vitro, but they just missed the statistical significance threshold when rho was used as a bacterial reference gene. The expression pattern of pdeB was consistent among each biological replicate, however. In full transparency, pdeB qPCR was originally performed with recA as a reference standard (bioRxiv preprint, ver 1). Here, significant changes in pdeB expression were observed in the in vitro assays comparing WT and ΔSfPat. These results prompted us to study changes in pdeB expression during in vivo colonization experiments, which also revealed significant changes. However, there was a concern that a potential SOS response would also activate recA, despite our preliminary data suggesting SOS was not involved. As a precautionary, we repeated the experiments with rho as a reference gene after it was identified as a stable reference. However, with rho as a reference gene, statistically significant responses were noted during in vivo colonization, but not in the in vitro assays.

In the current manuscript, one prophage was targeted based on preliminary findings indicating that the SfPat prophage region influences behaviors likely to impact colonization of the Ciona robusta gut. A separate genetic segment was also previously targeted for deletion as a misidentified prophage-like region, but that strain is not included in the current description. The currently presented data indicate that the observed phenomena can be attributed to the SfPat prophage.

Regarding the microbe-host interaction, it is not clear why the increased retention ability of the prophage deletion strain did not lead to greater cell retention after 24 hours, especially since no differences in the immune response were observed at that time point.

A predominantly adherent (non-motile) phenotype would likely facilitate elimination within fecal strings. There is substantial evidence from multiple model systems that strong swimming ability enhances the exploration and colonization of mucosal surfaces. Swimming helps with the penetration of mucus layers, chemotaxis toward epithelial surfaces, and overall “decision-making” in terms of shifting from a free-swimming (planktonic) state in the lumen within dietary material to a more sessile, adherent phenotype at the mucosal surface.

Concerning the methodological approach, I am puzzled as to why the authors opted for qPCR instead of transcriptomics or proteomics. The latter approaches could have provided a broader understanding of the prophage's impact on both the microbe and the host.

We agree with the reviewer that a transcriptomics approach would provide a broader understanding of the prophage’s impact on the microbe and animal host. Future studies will include a full multi-omic evaluation of this interaction.

**Reviewer #1 (Recommendations for the authors):**
Besides my above mentioned issues, I have a few more mini things:(A) what makes S. fidelis being a persistant member of the host microbiome? Please elaborate more on quantitive studies in this respect. –

Shewanella species are stable members of the Ciona gut, and previous efforts (Dishaw et al, 2016) revealed that chitin and/or secreted host effectors could influence biofilm formation. The Ciona gut produces copious amounts of endogenous chitin-rich mucus, and a variety of bacteria have been identified that thrive under these conditions. In addition, versatile bacteria like Shewanella sp. likely expand the metabolic potential of filter-feeders like Ciona. Thus, our subsequent studies began to focus on these and other microbes isolated from the Ciona gut that appear to be stable residents. Identical strains have been recovered numerous times (since 2011) from this wild population of Ciona robusta.

(B) The authors use the word inter kingdom and refer to phage, bacterium and animal. As phages are not part of the three kingdoms of life I believe the terminology is wrong.

Thank you for bringing this to our attention. In this context, we were referring to bacteria+phage as a unit and their interkingdom interaction with the animal host. But we recognize that this term can be misleading. Another, more appropriate term is ‘tripartite,’ and we have changed interkingdom to tripartite as appropriate, e.g., the abstract.

(C) I like lines 55-61 and was expecting to see in the manuscript what of those things would be true for the chosen prophage.

We looked at the coding region annotations within the prophage and the adjacent regions. The prophage coding regions are mostly annotated as unknown or predicted proteins, and a few as known phage-related components. We intend to reanalyze future and improved annotations and conduct deletion experiments targeting specific open reading frames (ORFs).

(D) In line 76 the authors mention a Gödecke reference for Pseudomonas. I believe that this paper only deals with S. oneidensis.

The inadvertent Gödecke reference has been removed.

(E) All figures: The captions are too short to understand what the figures are showing and everything is too small and hard to read or see. Along these lines it is often unclear what the many datapoints show. Biological replicates, technical replicates....Overall figure 1 does not seem to contain much information.

Figures and captions have been improved as suggested. Thank you for bringing this to our attention.

(F) Figure 3 what are a and b showing?

Figure and descriptive legend have been improved.

(G) Figure 4: Why did the author check expression only for one gene after 1 h but several genes after 24 h?

Since we observed that in vitro VCBP-C alters biofilms of S. fidelis 3313 (Dishaw et al 2016), we hypothesized that the bacteria may alter host VCBP-C expression and that the influence of integrated prophages may further modulate gene expression. Since VCBP-C is endogenously expressed in the gut of Ciona, we expected that early exposure/colonization (one hour) would be crucial for the bacterial-VCBP interactions. Hence, the VCBP-C was our primary target. We then tested multiple immune response genes at 24 hours to get a more detailed understanding of the maturing immune responses. Future studies will expand our efforts using global transcriptomics to understand better the immune response during bacterial exposure and colonization events.

(H) Do the authors mean stationary or localised?

We are not sure about the context of the reviewer’s question here but we think our modifications have addressed these concerns.

**Reviewer #2 (Public review):**
Summary:In the manuscript, "Prophage regulation of Shewanella fidelis 3313 motility and biofilm formation: implications for gut colonization dynamics in Ciona robusta", the authors are experimentally investigating the idea that integrated viruses (prophages) within a bacterial colonizer of the host Ciona robusta affect both the colonizer and the host. They found a prophage within the Ciona robusta colonizing bacterium Shewanella fidelis 3313, which affected both the bacteria and host. This prophage does so by regulating the phosphodiesterase gene pdeB in the bacterium when the bacterium has colonized the host. The prophage also regulates the activity of the host immune gene VCBP-C during early bacterial colonization. Prophage effects on both these genes affect the precise localization of the colonizing bacterium, motility of the bacterium, and bacterial biofilm formation on the host. Interestingly, VCBP-C expression also suppressed a prophage structural protein, creating a tripartite feedback loop in this symbiosis. This is exciting research that adds to the emerging body of evidence that prophages can have beneficial effects not only on their host bacteria but also on how that bacteria interacts in its environment. This study establishes the evolutionary conservation of this concept with intriguing implications of prophage effects on tripartite interactions.Strengths:This research effectively shows that a prophage within a bacterium colonizing a model ascidian affects both the bacterium and the host in vivo. These data establish the prophage effects on bacterial activity and expand these effects to the natural interactions within the host animal. The effects of the prophage through deletion on a suite of host genes are a strength, as shown by striking microscopy.Weaknesses:Unfortunately, there are abundant negative data that cast some limitations on the interpretation of the data. That is, examining specific gene expression has its limitations, which could be avoided by global transcriptomics of the bacteria and the host during colonization by the prophage-containing and prophage-deleted bacteria (1 hour and 24 hours). In this way, the tripartite interactions leading to mechanism could be better established.

We thank the reviewer for their comments and recognize this important limitation. As a follow-up to the current study, we plan to perform more comprehensive global meta-transcriptomics analyses to better understand differentially expressed genes across both the host and microbe during colonization.

Impact:The authors are correct to speculate that this research can have a significant impact on many animal microbiome studies, since bacterial lysogens are prevalent in most microbiomes. Screening for prophages, determining whether they are active, and "curing" the host bacteria of active prophages are effective tools for understanding the effects these mobile elements have on microbiomes. There are many potential effects of these elements in vivo, both positive and negative, this research is a good example of why this research should be explored.Context:The research area of prophage effects on host bacteria in vitro has been studied for decades, while these interactions in combination with animal hosts in vivo have been recent. The significance of this research shows that there could be divergent effects based on whether the study is conducted in vitro or in vivo. The in vivo results were striking. This is particularly so with the microscopy images. The benefit of using Ciona is that it has a translucent body which allows for following microbial localization. This is in contrast to mammalian studies where following microbial localization would either be difficult or near impossible.
**Reviewer #2 (Recommendations for the authors):**
In general, I found that the research shown in this manuscript is solid, and the manuscript is well-written. I have no specific comments about the writing of the manuscript that would be of benefit.Figure 1 would benefit from the shrinking of white space between panels a and b. Also, in panel b, it is very difficult to read the x-axis, the number of basepairs. It is suggested to increase this font size.

Figure 1 has been improved as suggested.

Figure 2 is fine, however, what do three asterisks (***) in panel a signify? It is not described in the legend. One minor point that affects data understanding as presented, the wildtype (WT) change in expression is normalized to itself, therefore always equaling 1.0. This method of presentation muddies the variation in gene expression in the presence of the prophage. This is not an issue in Figure 2, but does have an effect on understanding Figure 2 - figure supplement 1.Figure 2 - figure supplement 1, as stated above, the normalization of the WT change in gene expression to 1.0 makes it difficult to understand the results. Why is pilZ change in gene expression not significant in panel s1a? It seems the median change is 50%, or whatever averaging is done, it's unclear whether this is the median and whether the error bars are standard deviation or some other metric.These should be defined in the statistical analysis section of the methods or in the legend itself. Further, in panel s1b, why is the reduction in gene expression of pdeB statistically significant, while a similar reduction in gene expression of pleD is not statistically significant?

RQ values were calculated from 2^-ddCt^. The error bars in the figures were calculated by adding or subtracting the standard error from RQ. Since WT was used as a reference value for qPCR, the RQ value was normalized as 1 for all replicates and nonparametric tests were used to calculate the statistical significance. The values for pilZ were very close to significant; a value of 0.063 was derived via the Wilcoxon test. Only the changes in expression of pdeB were determined to be statistically significant, via the Wilcoxon test.

Figure 3 panels a and b would be helped by having the same y-axis for each. It is impressive the amount of WT bacterial colonization takes place in 24 hours, particularly in the absence of the prophage, but it does not appear as impressive when the axes are changed between panels. Similar axes should be considered for every comparative graph.Figure 3 - figure supplement 1 legend would benefit from the same description of the animal's digestive locations as in the legend in Figure 3.

We appreciate these suggestions and have made these changes accordingly. We have remade and combined Figure 3 a and b

Figure 4, while it is unfortunate that none of the immune genes evaluated had a response to the deletion of the SfPat prophage in S. fidelis 3313 at 24 hours, did any of these genes have an effect at 1 hour of evaluation as VCBP-C did?

The expression of this expanded gene set was not evaluated at one hour. This time point will, however, be included in our global evaluation of gene expression in our future transcriptome sequencing effort.

Figure 5, the only question I have with these data is whether or not there is a dose-dependent effect of VCBP-C on SfPat P5 expression?

Prior studies have found VCBP-C can impact biofilm formation in Shewanella sp. in a dose-dependent manner (some of the data appears in Dishaw et al, 2016). However, we have not yet considered whether VCBP-C impacts the expression of SfPat P5 (a phage capsid component) in a dose-dependent manner. We will consider this in future experimental designs.

It is mentioned in the introduction (and data shown in the preprint) that there is more than one active prophage in Shewanella fidelis 3313. The preprint data shows that the Mu prophages had little effect on the studies. It may be worth discussing the presence and lack of effects of these Mu prophages. It also may lead to some discussion about the complexities of polylysogeny (as discussed by Silpe, et al, Nature, 2023).

A full-length, inducible, Mu-like prophage region has been identified in the genome that has not been targeted for deletion, but will be included in follow-up studies. An earlier incomplete genome assembly contributed to the incorrect targeting and deletion of a prior Mu-like region, which was discussed in an earlier preprint version. Discussion and references to that strain have been removed from the more recent preprint versions. For clarity, the current manuscript describes strains that remain focused on the SfPat prophage, noting its contribution to the observed behavioral changes / traits.

Is there any spontaneous induction of SfPat in vitro or in vivo with temperature change (prophages have been induced with heat stress), excessive UV exposure, or mitomycin C treatment?

Preliminary induction studies using UV, mitomycin C, and temperature have been completed, but remain inconclusive with SfPat due to inconsistent induction patterns.

Could you speculate, or perhaps do the experiment, as to whether the addition of VCBP-C to S. fidelis 3313 cultures affects biofilm production? The deletion of SfPat leads to greater biofilm production in vitro, while exogenously added VCBP-C represses SfPat P5 expression, would VCPB-C addition lead to greater biofilm production? Lastly, and this may be a failure of my understanding, is VCBP-C able to bind to S. fidelis? If so, does the prophage alter the bacteria and, consequently, the ability of VCBP-C to bind to the bacteria?

Our lab is actively working to better understand the physical interactions of VCBP-C and bacteria, particularly lysogenic bacteria. Deletion mutants are helping us better understand the potential influence of the bacterial accessory genome on interactions with host immune mediators. Biofilm assays have been done in the context of VCBP-C (Dishaw et al, 2016). Subsequently, we tested the influence of 50 µg/ml VCBP-C on WT and prophage KO-strains, which include SfPat KO along with neutral (control) regions of the genome. We found that the presence of VCBP-C reduced biofilm formation in WT and phage KO variants at 4 hrs and 24 hrs. However, at 12 hrs, VCBP-C treatment appears to increase biofilm formation in the phage-KO strain. While the role (if any) of SfMu is remains unclear, these preliminary data imply the existence of a feedback circuit (influenced by time) where immune effector binding and prophage influence on host gene expression together shape retention outcomes in the gut microbiome. This hypothesis remains to be tested further.

**Author response image 1. sa4fig1:** WT S. fidelis 3313 was exposed in vitro to 50 µg/ml VCBP-C in stationary cultures. Biofilms were observed for 24hrs. At 12 hrs, the presence of VCBP-C increased the amount of biofilms, whereas reduced biofilms were observed at 4 and 24hrs. Our findings (manuscript Fig 2a) reveal that SfPat contributes to biofilm formation, exposure to SfPat deletion mutants increases host VCBP-C expression (manuscript Fig. 4a), and VCBP-C binding to WT S. fidelis 3313 reduces the expression of SfPat P5 capsid protein (manuscript Fig. 5). These findings suggest that in vivo exposure/ colonization assays benefit from detailed time-course observations to be further explored in follow-up, future experiments.

**Reviewer #3 (Public review):**
In this manuscript, Natarajan and colleagues report on the role of a prophage, termed SfPat, in the regulation of motility and biofilm formation by the marine bacterium Shewanella fidelis. The authors investigate the in vivo relevance of prophage carriage by studying the gut occupation patterns of Shewanella fidelis wild-type and an isogenic SfPat- mutant derivative in a model organism, juveniles of the marine tunicate Ciona robusta. The role of bacterial prophages in regulating bacterial lifestyle adaptation and niche occupation is a relatively underexplored field, and efforts in this direction are appreciated.While the research question is interesting, the work presented lacks clarity in its support for several major claims, and, at times, the authors do not adequately explain their data.Major concerns:(1) Prophage deletion renders the SfPat- mutant derivative substantially less motile and with a higher biofilm formation capacity than the WT (Fig. 2a-b). The authors claim the mutant is otherwise isogenic to the WT strain upon sequence comparison of draft genome sequences (I'll take the opportunity to comment here that GenBank accessions are preferable to BioSample accessions in Table 1). Even in the absence of secondary mutations, complementation is needed to validate functional associations (i.e., phenotype restoration). A strategy for this could be phage reintegration into the mutant strain (PMID: 19005496).

We are currently investigating complementation strategies. However, there have been some challenges in re-infecting and/or reintegrating the prophage into the genome. A preferred integration site may be damaged due to the deletion approach. While the SfPat prophage has mostly predicted genes of unknown function or significance, we have begun prioritizing the deletion of distinct segments to help identify functional relevance.

(2) The authors claim that the downshift in motility (concomitant with an upshift in biofilm formation) is likely mediated by the activity of c-di-GMP turnover proteins. Specifically, the authors point to the c-di-GMP-specific phosphodiesterase PdeB as a key mediator, after finding lower transcript levels for its coding gene in vivo (lines 148-151, Fig. 2c), and suggesting higher activity of this protein in live animals (!)(line 229). I have several concerns here:(2.1) Findings shown in Fig. 2a-b are in vitro, yet no altered transcript levels for pdeB were recorded (Fig. 2c). Why do the authors base their inferences only on in vivo data?(2.2) Somewhat altered transcript levels alone are insufficient for making associations, let alone solid statements. Often, the activity of c-di-GMP turnover proteins is local and/or depends on the activation of specific sensory modules - in the case of PdeB, a PAS domain and a periplasmic sensor domain (PMID: 35501424). This has not been explored in the manuscript, i.e., specific activation vs. global alterations of cellular c-di-GMP pools (or involvement of other proteins, please see below). Additional experiments are needed to confirm the involvement of PdeB. Gaining such mechanistic insights would greatly enhance the impact of this study.(2.3) What is the rationale behind selecting only four genes to probe the influence of the prophage on Ciona gut colonization by determining their transcript levels in vitro and in vivo? If the authors attribute the distinct behavior of the mutant to altered c-di-GMP homeostasis, as may be plausible, why did the authors choose those four genes specifically and not, for example, the many other c-di-GMP turnover protein-coding genes or c-di-GMP effectors present in the S. fidelis genome? This methodological approach seems inadequate to me, and the conclusions on the potential implication of PdeB are premature.

We chose to study genes that were shown previously to influence biofilms and motility in a cyclic-di-GMP dependent manner in a Shewanella spp (Chao et al 2013, S Rakshe 2011). Future transcriptomic efforts and targeted deletion approaches will further define the specific influence of prophages.

(3) The behavior of the WT strain and the prophage deletion mutant is insufficiently characterized. For instance, how do the authors know that the higher retention capacity reported for the WT strain with respect to the mutant (Fig. 3b) is not merely a consequence of, e.g., a higher growth rate? It would be worth investigating this further, ideally under conditions reflecting the host environment.

To clarify the method, in vitro growth curves did not suggest any significant difference in growth rate between the WT and the deletion mutant strains. Subsequently, for the in vivo experiments, bacterial cultures were pelleted and resuspended in sterile, nutrient-free artificial seawater. This limits growth until the bacterial strains are introduced to the animals.

(4) Related to the above, sometimes the authors refer to "retention" (e.g., line 162) and at other instances to "colonization" (e.g., line 161), or even adhesion (line 225). These are distinct processes. The authors have only tracked the presence of bacteria by fluorescence labeling; adhesion or colonization has not been assessed or demonstrated in vivo. Please revise.

We thank the reviewer for this feedback; the manuscript has been revised accordingly. While we refer to our assays as ‘colonization assays,’ we report results of ‘retention’ of various bacterial strains in the ‘exposed’ animals. Furthermore, when fluorescent staining is utilized, we report retention in defined niches. Since colonization is likely a two-step process, i.e., (1) retention and (2) colonization or long-term establishment of these microbial communities, using these terms correctly is warranted. In separate (unpublished) surveys of adult animals taken from the field, identical strains have been recovered numerous times over a twelve-year period.

(5) The higher CFU numbers for the WT after 24 h (line 161) might also indicate a role of motility for successful niche occupation or dissemination in vivo. The authors could test this hypothesis by examining the behavior of, e.g., flagellar mutants in their in vivo model.

Interestingly, we find numerous flagellar/motility-associated protein coding genes like Flg, Fli and Fle present within the S. fidelis genome possessing an EAL domain, implicating them in the regulation of cyclic-di-GMP. Hence, a future global transcriptomic approach will help improve our understanding of the roles of these regulatory pathways.

(6) The endpoint of experiments with a mixed WT-mutant inoculum (assumedly 1:1? Please specify) was set to 1 h, I assume because of the differences observed in CFU counts after 24 h. In vivo findings shown in Fig. 3c-e are, prima facie, somewhat contradictory. The authors report preferential occupation of the esophagus by the WT (line 223), which seems proficient from evidence shown in Fig. S3. Yet, there is marginal presence of the WT in the esophagus in experiments with a mixed inoculum (Fig. 3d) or none at all (Fig. 3e). Likewise, the authors claim preferential "adhesion to stomach folds" by the mutant strain (line 225), but this is not evident from Fig. 3e. In fact, the occupation patterns by the WT and mutant strain in the stomach in panel 3e appear to differ from what is shown in panel 3d. The same holds true for the claimed "preferential localization of the WT in the pyloric cecum," with Fig. 3d showing a yellow signal that indicates the coexistence of WT and mutant.

The results section is reworded to improve clarity. The WT and KO are mixed 1:1 to achieve the 10^7^ cfu count.

(7) In general, and especially for in vivo data, there is considerable variability that precludes drawing conclusions beyond mere trends. One could attribute such variability in vivo to the employed model organism (which is not germ-free), differences between individuals, and other factors. This should be discussed more openly in the main text and presented as a limitation of the study.

Yes, a salient feature of this model is that we can leverage genetic diversity in our experimental design, but it can introduce experimental variability.

Even with such intrinsic factors affecting in vivo measurements, certain in vitro experiments, which are expected, in principle, to yield more reproducible results, also show high variability (e.g., Fig. 5). What do the authors attribute this variability to?

For experiments involving VCBP-C protein, we can use affinity-purified protein recovered from live animals, or recombinant protein that we synthesize in-house (Dishaw et al 2011, 2016). In the latter, we often observe slight lot-to-lot variation in affinity for the target (the bacterial surface). To account for this variation and to ensure the observations are robust despite it, production lots can be mixed in additional biological replicates. As such, slight variability in the in vitro assays can be due to this batch effect.

(8) Line 198-199: Why not look for potential prophage excision directly rather than relying on indirect, presumptive evidence based on qPCR?

The decision to rely on qPCR of prophage structural genes was based on preliminary data, in particular among lysogens possessing more than one prophage. Neither the plaque assay nor SYBR Gold staining could distinguish among the particles, and TEM imaging was not sufficiently qualitative. Since these prophages do not exclusively produce particles when induced, qPCR targeting structural proteins was found to be most informative.

**Reviewer #3 (Recommendations for the authors):**
Other major comments:Line 137 (and Fig. 2 legend): The authors did not test chemotaxis towards any specific chemoeffector, only motility. Please correct and see below my comments about motility assays.

The reviewer is correct; we have modified our descriptors.

Lines 142-144: The authors conflate quorum sensing with c-di-GMP metabolism. If the authors measured the expression of genes "regulating cyclic di-GMP," it is likely because c-di-GMP is known to regulate the switch between planktonic and sessile lifestyles. However, whether this is mediated by quorum sensing is a separate issue that was not explored in this work. Please revise.

Thank you; these changes were made accordingly.

Line 150: c-di-GMP is not a quorum sensing signal; please correct.

Yes, we corrected the inadvertent yet misleading statement.

Line 193: Please clarify "RNA was extracted from the biofilms." If S. fidelis was grown on "MA [Marine Agar] for 24 h in the presence or absence of 50 µg/ml VCBP-C" (lines 192-193), was RNA isolated from colonies growing on the plates? Was VCBP-C added to the agar? This is also unclear in the Methods section (lines 381-384), where it seems the authors conducted this experiment using broth cultures in multiwell plates, removing the supernatant, and extracting RNA from the biofilms (i.e., cells adhered to the walls and bottom of the wells?). Why only biofilm cells?

Thank you for bringing this to our attention. We have rewritten the appropriate sections and methods to improve clarity. Following our initial studies, which revealed differential bacterial phenotypes (biofilm formation and motility assays), we decided to target and investigate gene expression in the biofilms. This way, the sessile cells that were not part of the biofilm do not obfuscate the data.

Lines 204-205: The authors should refer to the behavior of the mutant, since they did not test what happens upon prophage integration, but after prophage deletion.

The wording has been changed accordingly.

Lines 206-207: Please explain why the authors state that "these different bacterial phenotypes" (referring to altered biofilm formation and motility) "influence host immune responses in a manner consistent with influences on gut colonization dynamics". What specific relationship are the authors suggesting between these processes, and in what way is this "consistent"?

We previously demonstrated (Dishaw et al 2016) that copious amounts of VCBP-C protein are present under normal conditions in the gut and mostly found tethered to chitin-rich mucus lining the gut epithelium. The up-regulation of VCBP-C within one hour of exposure to the SfPat mutant relative to the WT S. fidelis is consistent with a role for VCBP-C in modulating bacterial settlement dynamics (Dishaw et al 2016). The mutant phenotype of reduced swimming and increased biofilm production is a likely trigger for the increased production of this secreted immune effector that may influence the retention of this bacterial variant, relative to the WT.

Line 229: Apart from what I noted above about the authors' claim regarding PdeB activity, I believe the figure referred to here should be Fig. 2, not Fig. 5.

Thank you for catching that oversight. It has been corrected.

Figure 1: Was hypothetical protein 2 included in the deletion?

Yes, the hypothetical protein 2 was included in the deletion

Figure 3a-b: It is challenging to interpret data on plots using so many colors - including what appears to be a white circle (?) in Fig. 3a. How many replicates are represented here? Is it indeed n=3 in Fig. 3a and n=6 in Fig. 3b?

Figure 3a is a bee swarm plot. Each color represents biological replicates, and the smaller circles represent technical replicates. It facilitates showing ALL the data, including the spread of the data. Regarding the number replicates, 3a and 3b are different experiments, with 3a representing a biofilm assay with three biological replicates and 3b a motility assay with six biological replicates.

Figure 3: An explanation for the abbreviation "FP" is missing.

Thank you for catching this oversight. The abbreviation has been defined.

Figure S3: FP, which is proficiently occupied by the WT strain (Fig. S3a), is not labeled in the images provided for the mutant (Fig. S3c-d). It would be helpful to show it for comparison.

Those other images did not have fecal pellets to label; however, Figure 3c does show a fecal pellet for an animal exposed to both WT and the SfPat mutant.

Questions and comments regarding methods:Lines 290-291, 307: Please indicate an approximate range for "room temperature."

The information has been added to the revised manuscript.

Lines 292, 302: Why use hybrid LB/MB broth and agar? And strictly speaking, which LB formula (Lennox/Luria/Miller)?

The hybrid broth reduces the concentration of salts that can interfere in some assays. The LB formula was Luria, and it is now included in the manuscript.

Lines 300-302: The conjugation procedure is poorly described. It seems the authors conducted conjugal transfer by biparental mating in broth culture by inoculating a single colony of S. fidelis 3313 into an already grown culture of the *E. coli* donor strain?

The biparental mating was done on plates; the manuscript has been clarified.

Motility assay concerns:Swimming motility is generally assayed in soft agar (0.25-0.3% w/v). Why did the authors use 0.5% low-melt agarose? Usually, agar is employed instead of agarose, and such a high concentration of solidifying agent typically prevents proper swimming (see e.g. Kearns 2010).

Our laboratory uses low-melt agarose for phage propagation and other assays. We continued using it because we observed robust and reproducible results in the swarming and swimming motility assays. In addition, 0.5% agarose is less dense than 0.5% agar, and its consistency is similar to that of the lower percentage soft agar.

Lines 316-317: Please clarify: what is the "overlay motility assay" that was carried out "overnight at RT and then inoculated onto the center of soft agar"? Was this a two-step experiment? How were bacteria inoculated (stabbed, injected)? If injected, what volume and cell density were used?

Thank you for bringing this to our attention. The methods section has been revised for clarity.

Line 319: Each variable tested in duplicate? From what I understand, the only variable measured in this test is the diameter of the swimming halos. Do the authors mean they used two biological replicates? If so, please indicate the number of technical replicates as well.

Multiple biological replicates were performed, each time with two technical replicates. Two perpendicular measurements (of diameter) for each technical replicate was recorded to avoid bias. The methods section has been edited to improve clarity.

Line 320: Were the swimming halos asymmetrical, hence the need to take two perpendicular measurements? If that was the case, it could indicate an excessive amount of solidifying agent.

The halos were sometimes asymmetric, but to avoid variation across datasets, it became standard practice to measure perpendicular distances as stated above.

Regarding qPCR experiments:Please clarify how normalization of transcript levels was performed.It seems the authors conducted a double normalization, first with respect to the calibrator (rho), and again using the wild-type as a baseline reference for fold-change calculations (absence of error bars for WT data). If so, please specify on the vertical axes of the figures and in the Methods/figure legends.Since, in addition to rho, the authors assessed the expression stability of the "housekeeping" genes gyrB and recA, please also include the primers used for these genes.

The appropriate manuscript sections have been updated for clarity. The bacterial qPCR was normalized to an internal standard, and then relative expression differences between SfPat and the WT were determined. The missing primer sequences have also been added.

Observations:Figure 2a-b: It is intriguing that the remarkable reduction in motility of the mutant is not associated with a comparably significant increase in biofilm formation.

A statistically significant increase in biofilm was observed, along with a decrease in motility. As is common in crystal violet assays, some of the tertiary structures were not very stable and likely washed out during processing.

Additionally, it is noteworthy that data for the mutant in panel 2a exhibit minimal variability, with all OD570 recordings being around 3.0. Did the authors dilute the crystal violet elution solution after adding acetic acid, or might they have reached the saturation limit of the spectrophotometer?

The eluted acetic acid was not diluted further, and significant changes were observed. If the solution had been further diluted, the observed changes might have been more pronounced.

Minor comments and recommendations:

All the suggested changes below have been incorporated

• Line 55: "Antibiotic resistance determinants" might be preferable to "genes" to avoid using "genes" twice in the same sentence.

• Line 75-76: Italicize *Pseudomonas aeruginosa*.

• Line 134: Instead of "at least," specify the average fold-change.

• Line 141: In the heading, refer to the influence of the "prophage" (singular) rather than "prophages" (plural).

• Discussion (style): Consider using past tense for phrases like "we utilize..." (line 202); "we find..." (line 204), etc.

• Line 365 and elsewhere: Consider "mRNA levels" or "transcript levels" instead of "gene expression".

• Table 3: UQ950 is a strain, not a plasmid. I assume the plasmid carried by UQ950 is pSMV3.